# Effects of Critical Thinking Disposition, Clinical Judgement, and Nurse–Physician Collaboration on Triage Competency Among Triage Nurses

**DOI:** 10.3390/healthcare13040405

**Published:** 2025-02-13

**Authors:** Ji-Won Song, Hyung-Ran Park

**Affiliations:** 1Chungbuk National University Hospital, 776, Cheongju 28644, Chungbuk, Republic of Korea; sa9766@naver.com; 2Department of Nursing Science, Research Institute of Nursing Science, College of Medicine, Chungbuk National University, Cheongju 28644, Chungbuk, Republic of Korea

**Keywords:** critical thinking disposition, clinical judgement, nurse–physician collaboration, triage competency, triage nurse

## Abstract

**Background/Objectives**: This study aimed to investigate the relationship between critical thinking disposition, clinical judgement, nurse–physician collaboration, and triage competency among triage nurses and to identify the factors influencing triage competency. **Methods**: This descriptive survey study included 152 triage nurses from one tertiary and six general hospitals. Data were collected from 1 September to 1 November 2023 and analysed using SPSS 29.0. **Results**: Triage competency was positively correlated with critical thinking disposition (*r* = 0.55, *p* < 0.001) and clinical judgement (*r* = 0.61, *p* < 0.001), while it was negatively correlated with nurse–physician collaboration (*r* = −0.52, *p* < 0.001). The hierarchical regression analysis showed that five variables, critical thinking disposition, clinical judgement, nurse–physician collaboration, education level, and years of experience in the emergency department, significantly influenced triage competency. The explanatory power of these variables for triage competency was 63.7% (F = 27.52, *p* < 0.001). **Conclusions**: Based on the findings, increasing the level of cooperation and fostering more cooperative relationships between nurses and physicians are necessary measures to improve their triage competency.

## 1. Introduction

In emergency medical centres, triage is essential for the efficient allocation of limited healthcare resources to patients and for the systematic provision of emergency care [1]. Triage nurses are healthcare providers who classify critically ill patients according to their signs and symptoms [2]. When performing triage, a lack of competency can cause errors in setting priorities, resulting in inefficient use of resources [2]. Given that the severities determined by nurses significantly influence patients’ treatment and prognosis, developing sufficient triage competency is vital for nurses [3,4]. Triage competency has been suggested to directly affect nursing competencies trained in the emergency department through expert evaluation of patients’ urgency and the efficient management and allocation of medical resources [5]. In previous studies on factors related to triage competency, nurse-related factors included critical thinking disposition [6,7,8] and clinical judgement [3,8], while environmental factors included nurse–physician collaboration [9] and stress [10,11].

Critical thinking is objective and cognitive judgement based on analysis and inference in a situation to solve a problem [12]. Critical thinking disposition refers to consistent internal motivation and an active attitude towards critical thinking [8]. In clinical situations, nurses must demonstrate competency in evidence-based critical thinking to solve patients’ health problems [7]. In previous studies, critical thinking disposition has been shown to improve nursing practice [13], job performance [14], and triage [15]. In triage, nurses need to practice rapidly and accurately triaging patients with complex health problems [2], and this can be achieved through critical thinking [14]. Therefore, investigating the relationship between triage competency and critical thinking disposition in triage nurses is necessary.

Although clinical judgement has previously been used interchangeably with critical thinking, they are two distinct concepts [16,17]. Clinical judgement is a reflective and inferential process of using available data, based on extensive knowledge, to reach a clinical conclusion [16]. In the emergency room, where unpredictable incidents often occur, triage needs to be performed promptly with limited resources when the number of patients increases. This can lead to reduced accuracy, potentially harming patient safety or satisfaction [2,18,19]. Clinical judgement enables nurses to provide high-quality nursing by analysing a critical situation and appropriately deciding what knowledge and resources to use in triage [16]. In previous studies, appropriate clinical judgement, achieved by applying knowledge and clinical experience, was found to be essential for triage [18], and clinical judgement was suggested to be a professional capability and an essential clinical competency for triage nurses in the emergency department [19]. Thus, investigating the relationship between clinical judgement and triage competency in triage nurses is crucial.

In addition, nurse–physician collaboration is an environmental factor that promotes interactions among healthcare providers based on an atmosphere of respect and trust within the emergency healthcare system. It enables the efficient expression of knowledge and skills and is consequently closely related to triage competency [9,20]. In the Korean triage system, classifications are made by medical personnel, including physicians, nurses, and emergency medical technicians with triage qualifications [21,22]. Triage staff members are organised according to the medical institution; in most Korean hospitals, nurses are key staff members who either work independently or with a physician [21]. Nurses can exchange opinions with other members of the triage staff, such as other nurses or physicians, during the triage process. This could help reduce triage errors and ensure decisions are made quickly [21]. For triage nurses, teamwork and collaboration with other healthcare professionals are important clinical abilities that enable emergency patients to receive suitable treatment [9,23]. Nurses and physicians are the primary providers involved in patient treatment and effective cooperation between these professionals can ensure that nurses clearly and quickly interpret the condition of undiagnosed and unpredictable patients, both following physician instructions and effectively presenting their own opinions [9]. Previous studies have shown that effective nurse–physician cooperation can lead to optimal health outcomes for patients by enabling nurses to provide high-quality nursing care and improve patient safety [24]. Thus, research focusing on the effects of nurse–physician collaboration on triage competency among triage nurses is necessary.

This study aimed to investigate the levels of critical thinking disposition, clinical judgement, nurse–physician collaboration, and triage competency in triage nurses working in the emergency department and to determine the factors affecting triage competency among these nurses.

## 2. Materials and Methods

### 2.1. Design

This descriptive cross-sectional study aimed to determine the factors affecting triage competency among triage nurses working in the emergency department.

### 2.2. Participants and Setting

The participants in this study were triage nurses who had been working and performing triage for at least one year [25] in the emergency department of one tertiary hospital and six general hospitals in Chungcheong-do, Republic of Korea.

To calculate the required sample size, we used G*power 3.1.9.7 for multiple regression analysis, assuming a power of 0.90, a significance level of 0.05, a Cohen’s [26] moderate effect size of 0.15, and 11 predictive variables (eight general characteristics and three independent variables). Consequently, the required sample size was found to be 152 individuals. To account for a dropout rate of 10%, we distributed questionnaires to 167 individuals. We received 162 responses (97%), and after excluding 10 insincere responses, we included 152 individuals in the final analysis.

### 2.3. Measures

All instruments were used after obtaining permission from their authors.

#### 2.3.1. General and Triage-Related Characteristics

The general characteristics of the participants were sex, age, and education level. The triage-related characteristics were years of experience in the emergency department, months of triage experience, type of emergency medical centre, personnel involved in triage, and days worked in triage per week.

#### 2.3.2. Critical Thinking Disposition

Critical thinking disposition was measured using the critical thinking disposition instrument developed and validated by Yoon [27]. This instrument consists of 27 items, including intellectual passion and curiosity, prudence, healthy scepticism, self-confidence, intellectual fairness, objectivity, and systematicity. Each item is scored on a 5-point Likert scale, with higher scores indicating a stronger critical thinking disposition. In terms of the reliability of the instrument, Cronbach’s α was 0.84 at the time of development [27], 0.90 for triage nurses [28], and 0.84 in this study.

#### 2.3.3. Clinical Judgement

Clinical judgement was measured using the clinical judgement scale developed and validated by Kwon and Park [29]. This scale consists of 23 items, including integrated data analysis, intervention evaluation and reflections, intervention evidence, inter-professional collaboration, patient-centred nursing, and collaboration between nursing colleagues. Each item is scored on a 5-point Likert scale, with higher scores indicating better clinical judgement ability. Cronbach’s α was 0.91 at the time of development for nurses [29] and 0.92 in this study.

#### 2.3.4. Nurse–Physician Collaboration

Nurse–physician collaboration was measured using the Korean Version of the Nurse–Physician Collaboration Scale (K-NPCS), which was translated and validated by Mun [30] from the Nurse–Physician Collaboration Scale (NPCS) developed by Ushiro [31]. This scale consists of 27 items, including information sharing, decision-making about treatment and care, and nurse–physician relationships. Each item is scored on a 5-point Likert scale, with lower scores indicating better collaboration. In terms of reliability, Cronbach’s α was 0.8 for the original instrument in the study by Ushiro [31], 0.95 in the study by Mun [30], 0.92 for emergency nurses [9], and 0.96 in this study.

#### 2.3.5. Triage Competency

Triage competency was measured using the triage competency scale for emergency department nurses, which was developed and validated by Moon and Park [32]. This scale consists of 30 items, including clinical judgement, communication, professional identification, personal coping, and management. Each item is scored on a 5-point Likert scale, with possible scores ranging from 30 to 150 points, where higher scores indicate better triage competency. In terms of reliability, Cronbach’s α was 0.91 at the time of development [32], 0.91 for triage nurses [28] and 0.97 in this study.

### 2.4. Data Collection

The data for this study were collected between 1 September and 1 November 2023 after receiving approval from an institutional review board. We first obtained permission for data collection from the nursing and emergency department heads of the participating hospitals. The first author reached out to each emergency department and explained the study objectives and methods with written information. Triage nurses who voluntarily consented to participate signed informed consent forms and were provided with a questionnaire for data collection. To protect the privacy of the participants, data collection was conducted in a separate space, and the responses to the questionnaire were received in a sealed envelope; furthermore, an identification number was assigned in place of the participant’s name. A small token of gratitude (i.e., a toothbrush set worth 5000 KRW [3.43 USD]) was provided to the participants who completed the questionnaire.

### 2.5. Analysis

The collected data were analysed using descriptive statistics, independent *t*-tests, ANOVA, and Scheffé post-hoc test in SPSS Statistics 29.0. Pearson’s correlation coefficients were used to analyse the relationships among critical thinking disposition, clinical judgement, and nurse–physician collaboration. Hierarchical regression analysis was conducted to identify factors affecting the participants’ triage competency in order to consider their general characteristics.

### 2.6. Ethical Considerations

This study was reviewed and approved by an institutional review board at the researchers’ affiliated institution (2023-06-021-001). The participants received a written explanation of the study objectives and methods, and they voluntarily signed two written informed consent forms. One signed informed consent form was then provided to the participants. The explanation stated that participation in the study would involve no expected risks, that participants could withdraw at any time during the study, and that withdrawal would not result in any disadvantages. The participants were informed that the questionnaires would be assigned arbitrary identification numbers for management, would only be used for research purposes, and would be stored in a locked cabinet for data protection. In addition, they were informed that the privacy of their personal responses and information would be thoroughly maintained. The author collected the data either before or after each participant’s shift to avoid negatively affecting their work responsibilities.

## 3. Results

### 3.1. Differences in Triage Competency According to the Participants’ Characteristics

The mean age of the participants was 30.26 ± 6.14 years. Triage competency was significantly worse in those aged <30 years (F = 9.42, *p* < 0.001). It was significantly better for nurses with a master’s degree or higher (F = 12.51, *p* < 0.001) or for those with at least six years of experience in the emergency department (F = 17.37, *p* < 0.001). It was also higher in nurses with at least 12 months of triage experience (*t* = −4.16, *p* < 0.001; Table 1).

### 3.2. Descriptive Statistics of Critical Thinking Disposition, Clinical Judgement, and Nurse–Physician Collaboration on Triage Competency

The item mean critical thinking disposition score was 3.50 ± 0.36 (range 2.44–4.37), with an item mean score of 3.92 ± 0.47 (range 2.43–5.00) for clinical judgement. The item mean nurse–physician collaboration score was 2.29 ± 0.63 (range 1.00–4.00), and the item mean triage competency score was 4.00 ± 0.56 (2.47–5.00; Table 2).

### 3.3. Correlations Among Critical Thinking Disposition, Clinical Judgement, and Nurse–Physician Collaboration with Triage Competency

Triage competency was significantly positively correlated with critical thinking disposition (*r* = 0.55, *p* < 0.001) and clinical judgement (*r* = 0.61, *p* < 0.001) while being negatively correlated with nurse–physician collaboration (*r* = −0.52, *p* < 0.001; Table 3).

### 3.4. Factors Affecting Triage Competency

Multiple regression analysis using the hierarchical method was conducted to identify the factors influencing triage competency among the triage nurses. Critical thinking disposition, clinical judgement, and nurse–physician collaboration were entered as continuous independent variables. Categorical variables, including age, education level, experience in the emergency department, and triage experience, which showed significant differences in triage competency, were dummy-coded. Tolerance was above the cut-off value of 0.1 (range 0.44–0.69), the variance inflation factor was below the cut-off value of 10 (range 1.45–2.25), and the Durbin–Watson coefficient (1.948) was close to 2. This confirmed the absence of multicollinearity among the independent variables.

In Model 1, the participants’ general characteristics were used as control variables. These variables explained 25.5% (F = 8.40, *p* < 0.001) of the variance, and education level (≥Master) (β = 0.32, *p* = 0.001) and experience in the emergency department (≥6 years) (β = 0.34, *p* = 0.001) were influencing factors on triage competency. Model 2 was the result of the hierarchical entry of critical thinking disposition, clinical judgement, and nurse–physician collaboration (Table 4). The factors influencing triage competency were education level (≥Master) (β = 0.21, *p* = 0.003), experience in the emergency department (≥6 years) (β = 0.32, *p* < 0.001), critical thinking disposition (β = 0.27, *p* < 0.001), clinical judgement (β = 0.16, *p* = 0.008), and nurse–physician collaboration (β = −0.34, *p* < 0.001). These variables explained 63.7% of the variance in triage competency among the triage nurses (F = 27.52, *p* < 0.001; Figure 1).

## 4. Discussion

This study aimed to investigate the factors affecting triage competency in triage nurses. Consequently, nurse–physician collaboration, experience in the emergency department, critical thinking disposition, education level, and clinical judgement were found to affect triage competency in descending order. The regression analysis showed that these factors explained 63.7% of the variance in triage competency.

Nurse–physician collaboration was the factor that most significantly influenced triage competency. This finding is consistent with previous studies that reported an association between better nurse–physician collaboration and improved triage competency [9,33,34]. This is likely because collaboration with physicians allows nurses to better express their professional competencies and enhances the quality of their nursing practice [33]. However, the level of nurse–physician collaboration observed in our study was lower than that perceived by nurses in other departments [34]. This could be due to friction when coping with critical situations in the emergency department, conflicts arising from triage outcomes, or differing understandings of decisions when taking responsibility, all of which could explain the low perceived level of nurse–physician collaboration [35]. High patient severity and density in the emergency department create difficulties in effectively sharing opinions during triage [9,35]. Triage nurses work in a unique environment susceptible to interpersonal conflict due to the high severity of conditions and time constraints [2]. In this regard, given that mutual collaboration between physicians and nurses affects the work of nurses [33], promoting teamwork between healthcare professionals is crucial to improving triage competency [4] as a professional capability of triage nurses. The experience of nurse–physician collaboration in the emergency department must be investigated in future research.

In our study, stronger critical thinking disposition was associated with better triage competency, a finding similar to previous studies that reported a relationship between critical thinking disposition and improved nursing performance [13,14,28,36]. In the triage area in the emergency department, unexpected situations frequently occur. Critical thinking disposition can lead to better triage competency by enabling judgements based on objective criteria and evidence in situations with uncertain workloads [2,7]. This includes assessing patients’ general appearance and determining severity within seconds to prevent delays in subsequent treatment [15,28]. A previous study found that differences in critical thinking disposition could be investigated by implementing a web-based educational program to improve triage competency in triage nurses [6]. However, when designing such a program, detailed content for improving critical thinking disposition will have to be considered. In this study, clinical judgement was also identified as a significant factor influencing triage competency. In a previous study, emergency department triage nurses who had received web-based triage education showed better clinical judgement [6]. Clinical judgement is a core competency of nurses, enabling them to assess patients’ urgency and determine priorities for timely intervention. Therefore, continuous education programs using web-based methods are an essential piece of effectively improving clinical judgement.

Moreover, experience in the emergency department was a significant factor influencing triage competency in our study. This is consistent with previous studies that reported better triage competency in nurses [9,10,37]. To perform initial nursing tasks, including triage, at a professional level, nurses accumulate knowledge through experience caring for diverse emergency patients. This experience enhances their judgement and ability to react flexibly, ultimately improving triage competency [9,10]. In South Korea, at least one year of emergency department experience is required to perform triage, and only those with this experience can obtain the Korean Triage Acuity Scale qualification [9]. Korean triage nurses need systematic training management to improve their triage [35]. Ongoing professional development education is necessary for enhancing their skills [38]. Furthermore, to ensure expertise based on clinical experience for triage nurses in the emergency department, transfer to other departments should be restricted, and continuous education should be provided to improve triage accuracy [6]. We also found that a higher education level was associated with better triage competency. This is different from previous studies, which reported no difference in triage competency among nurses with a postgraduate degree [10,37]. Nurses who take a degree course to improve their expertise face new challenges, and in this process, they gain self-confidence and value their own competencies more highly [37]. Therefore, further research is needed to determine whether education level is a factor in improving triage competency so that active support and encouragement can be provided within nursing organisations.

This study aimed to investigate the levels of critical thinking disposition, clinical judgement, nurse–physician collaboration, and triage competency in emergency department nurses performing triage at regional and local emergency medical centres and to identify the factors affecting triage competency. Consequently, various factors were identified to directly influence triage competency. If validated and confirmed in future studies, our findings could help prepare institutions and establish systems to support nurse–physician collaboration and provide ongoing training to enhance triage competency among triage nurses in emergency departments.

### Study Limitations

Since this study only included triage nurses working in emergency departments in a certain region of Korea, the findings are limited in terms of their generalisability. Therefore, they should be validated with broader or different cultural populations across more diverse regions. Due to differences in the practices of triage staff nurses across emergency departments in different countries, a multicultural study should be considered to explore triage competency. In addition, we collected cross-sectional data using a self-reported survey; a long-term observation study should be considered. Moreover, we could not consider institutional characteristics, so the size and available resources of different emergency departments should be considered in future research. As a reward for participation, we gave each participant a toothbrush set worth 3.43 USD. While this incentive was small, it could have affected participation selection bias. Finally, as we recruited participants who worked in emergency departments through convenience sampling, randomisation should be considered to check for internal validity.

## 5. Conclusions

In this study, we investigated the relationships among critical thinking disposition, clinical judgement, nurse–physician collaboration, and triage competency of triage nurses. In our study, nurse–physician collaboration was confirmed to be the most important factor in triage nurses expressing their competencies. In addition to critical thinking disposition and clinical judgement, characteristics such as nurses’ experience in the emergency department and education level were also found to be important. These factors contribute to nurses’ triage expertise, which influences patients’ safety and the quality of care received in emergency departments. Our study has significant implications for nursing policy and practice to improve triage competency. It emphasises the need for organisational policies that foster collaborative relationships among healthcare professionals in triage settings. Furthermore, various applied educational interventions, such as web-based or smartphone-accessible training, should be implemented to strengthen critical thinking and clinical judgement. Continuing education opportunities at the postgraduate level should also be made available to further support the development of triage nurses.

## Figures and Tables

**Figure 1 healthcare-13-00405-f001:**
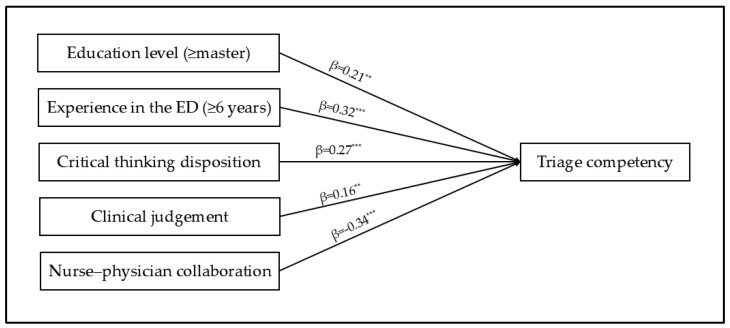
Factors affecting triage competency; ** *p* < 0.01, *** *p* < 0.001.

**Table 1 healthcare-13-00405-t001:** Differences in triage competency according to the participants’ characteristics (*N* = 152).

Variables	Categories	*n*	%	Triage Competency		
				M ± SD	*t*/F (*p*)	Scheffé
Sex	Male	38	25.0	117.97 ± 14.46	−0.52 (0.587)	
	Female	114	75.0	119.70 ± 17.69		
Age (years)	<30 ^a^	92	60.5	114.91 ± 15.91	9.42 (<0.001)	a < b,c
	30–<35 ^b^	39	25.7	124.00 ± 15.90		
	≥35 ^c^	21	13.8	129.57 ± 16.88		
Education level	Associate degree ^a^	16	10.5	120.19 ± 12.85	12.51 (<0.001)	a,b < c
	Bachelor’s degree ^b^	118	77.7	116.53 ± 16.66		
	≥Master’s degree ^c^	18	11.8	136.44 ± 10.92		
Experience in the ED (years)	<3 ^a^	60	39.5	111.92 ± 16.39	17.37 (<0.001)	a,b < c
	3–<6 ^b^	44	28.9	118.30 ± 16.00		
	≥6 ^c^	48	31.6	129.35 ± 13.20		
Triage experience (months)	<12	69	45.4	113.32 ± 15.72	−4.16 (<0.001)	
	≥12	83	54.6	124.22 ± 16.35		
Type of emergency medical centre	Regional emergency medical centre	56	36.8	116.27 ± 15.76	−1.68 (0.095)	
	Local emergency medical centre and other medical centre	96	63.2	121.02 ± 17.40		
Personnel involved in triage	Alone	43	28.3	123.98 ± 14.71	2.92 (0.570)	
	With a doctor	54	35.5	119.09 ± 18.80		
	With a nurse	55	36.2	115.76 ± 15.96		
Triage frequency (per week)	<3 days	61	40.1	118.85 ± 17.56	−0.25 (0.804)	
	≥3 days	91	59.9	119.55 ± 16.56		

Notes. n: frequency; ED: emergency department; M: mean; SD: standard deviation.

**Table 2 healthcare-13-00405-t002:** Levels of critical thinking disposition, clinical judgement, and nurse–physician collaboration on triage competency (*N* = 152).

Variables	Number of Items	Item Mean			Scale Range
		M ± SD	Min	Max	
Critical thinking disposition	27	3.50 ± 0.36	2.44	4.37	1–5
Clinical judgement	23	3.92 ± 0.47	2.43	5.00	1–5
Nurse–physician collaboration	27	2.29 ± 0.63	1.00	4.00	1–5
Triage competency	30	4.00 ± 0.56	2.47	5.00	1–5

Notes. M: mean; SD: standard deviation.

**Table 3 healthcare-13-00405-t003:** Correlations among critical thinking disposition, clinical judgement, and nurse–physician collaboration with triage competency (*N* = 152).

Variables	Critical Thinking Disposition	Clinical Judgement	Nurse–Physician Collaboration	Triage Competency
	*r* (*p*)			
Critical thinking disposition	1			
Clinical judgement	0.52 (<0.001)	1		
Nurse–physician collaboration	−0.26 (<0.001)	−0.56 (<0.001)	1	
Triage competency	0.55 (<0.001)	0.61 (<0.001)	−0.52 (<0.001)	1

**Table 4 healthcare-13-00405-t004:** Factors affecting triage competency (*N* = 152).

Variables	Model 1				Model 2			
	B	SE	β	*t* (*p*)	B	SE	β	*t* (*p*)
Constant	3.58	0.16		23.15 (<0.001)	1.97	0.42		4.65 (<0.001)
Age (30–<35 years) *	−0.01	0.12	−0.01	−0.06 (0.954)	0.03	0.08	0.03	0.42 (0.678)
Age (≥35 years) *	0.15	0.14	0.09	1.07 (0.286)	0.02	0.10	0.01	0.18 (0.858)
Education level (bachelor) *	0.08	0.14	0.06	0.60 (0.548)	0.08	0.10	0.06	0.85 (0.397)
Education level (≥master) *	0.55	0.17	0.32	3.28 (0.001)	0.37	0.12	0.21	3.04 (0.003)
Experience in the ED (3–<6 years) *	0.15	0.10	0.12	1.43 (0.156)	0.12	0.07	0.10	1.63 (0.106)
Experience in the ED (≥6 years) *	0.41	0.12	0.34	3.32 (0.001)	0.39	0.08	0.32	4.51 (<0.001)
Triage experience (≥12 months) *	0.13	0.10	0.11	1.32 (0.189)	0.08	0.07	0.07	1.20 (0.233)
Critical thinking disposition					0.43	0.10	0.27	4.48 (<0.001)
Clinical judgement					0.22	0.08	0.16	2.68 (0.008)
Nurse–physician collaboration					−0.31	0.05	−0.34	−5.65 (<0.001)
F (*p*)	8.40 (<0.001)				27.52 (<0.001)			
R^2^ (adjusted R^2^)	0.290 (0.255)				0.661 (0.637)			
R^2^ change					0.371			
Tolerance	0.46–0.70				0.44–0.69			
VIF	1.43–2.19				1.45–2.25			

Notes. B: unstandardised regression coefficient; ED: emergency department; SE: standard error; β: standardised regression coefficient; VIF: variance inflation factor; * Dummy variable: age (ref. < 30 years), education level (ref. = associate degree), experience in the ED (ref. < 3 years), triage experience (ref. < 12 months).

## Data Availability

Data is contained within the article.

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
