# Peer review of "Effects of Critical Thinking Disposition, Clinical Judgement, and Nurse–Physician Collaboration on Triage Competency Among Triage Nurses"

_healthcare, 2025, doi:10.3390/healthcare13040405_

Round 1

Reviewer 1 Report

Comments and Suggestions for Authors

I read with interest the paper from Song and Park. I believe that the manuscript does investigate an important topic, but the analysis does not give merit to the quality of the paper, potentially resuting in biased findings. This in turn weakens the robustness and generalizability of the conclusions. On the ohter hand, the manuscript has a strong theoretical background, good presentation, and good reproducibility due to the comprehensive usage of validated scales. While the study is relevant and addresses an important problem, significant revisions are required to meet the standards of high-quality research. Overall, I believe that the study has good potential, but in its current state its methodological shortcomings significantly diminish its contribution, and some major revisions are required.

Hereby I list my suggestions and concerns:

Introduction:

1) The Authors highlight the critical role of triage nurses in emergency departments, and identify key factors influencing triage competency, including critical thinking disposition, clinical judgment, and nurse–physician collaboration. However, there is little information regarding the study setting itself. As a reader, I'd be interested in some basic info on how Korean triage system as a whole works (i.e. if it follows always the same standard, if and where there is margin of discretion at the hospital/operator/patient level, etc.). 

2) Since the paper points at nurse-physician collaboration, some information on the triage process should also be given. It is my understanding that triage is a purely nursing activity, so the physician does not have any role in it. Since it may be (or not) the same in Korea, I believe that more clarity on the nurse-physician interaction that may happen in ED would add clarity to the paper.

Methods:

I have doubts on the overall data analysis of the study, since I believe the simple regression model proposed may not be the best choice; if not already done, I suggest the Auhtors may check with a statistician:

1) The Authors mention that the nurses in the sample were selected among seven general hospitals. This creates a nested data structure (nurses clustered within hospitals). It means that observations within the same hospital are expected to be more similar to each other than observations from different hospitals. Ignoring such clustering might lead to underestimated standard errors and inflated significance levels. A multi-level model would be more appropriate to account for the non-independence of observations.

2) Since one hospital may have contributed to many more participants than another, either in absolute numbers or in proportion, the data should be weighted to account for this imbalance.

Results

1) As per previous points, Cluster-level information should also be provided. Please consider that hospital-level characteristics (e.g., size, resource availability) may influence both the predictors (e.g., critical thinking) and outcomes (triage competency).

Discussion

1) The information on incentives for partecipation (A small token of gratitude was provided to the participants who completed the questionnaire) increases transparency. However, I believe Authors should evaluate the possibilty for this to introduce selection bias (maybe those who responde to incentive are the most/least collaborating?). If this is a possibility, it should be at least mentioned in discussion. In alternative, the Authors could analyze non-respondents characteristics, or describe how the incentive was offered to mitigate bias (e.g., equal opportunity for all eligible participants).

Author Response

Response to Reviewer 1 Comments

My co-author and I wish to re-submit our revised manuscript entitled “Effects of critical thinking disposition, clinical judgement, and nurse–physician collaboration on triage competency among triage nurses” with changes that thoroughly address your comments.

We thank you for your thoughtful suggestions and insights. The manuscript has benefited from your thorough feedback. We look forward to working with you to move this manuscript closer to publication in Healthcare.

The manuscript has been rechecked and the necessary changes have been made in accordance with your suggestions. The responses to all comments have been prepared and attached herewith/given below.

We appreciate your consideration.

Comment 1: Introduction: The Authors highlight the critical role of triage nurses in emergency departments, and identify key factors influencing triage competency, including critical thinking disposition, clinical judgment, and nurse–physician collaboration. However, there is little information regarding the study setting itself. As a reader, I'd be interested in some basic info on how Korean triage system as a whole works (i.e. if it follows always the same standard, if and where there is margin of discretion at the hospital/operator/patient level, etc.).

Response 1: Thank you for your careful review. We have added information regarding the Korean triage system in the Introduction (page 2, paragraph 3, lines 66–70).

“In the Korean triage system, classifications are made by medical personnel, including physicians, nurses, and emergency medical technicians with triage qualifications [21,22]. Triage staff members are organised according to the medical institution; in most Korean hospitals, nurses are key staff members who either work independently or with a physician [21].”

Comment 2: Introduction: Since the paper points at nurse-physician collaboration, some information on the triage process should also be given. It is my understanding that triage is a purely nursing activity, so the physician does not have any role in it. Since it may be (or not) the same in Korea, I believe that more clarity on the nurse-physician interaction that may happen in ED would add clarity to the paper.

Response 2: Thank you for your detailed comments. Based on your recommendation, we added information about the nurse–physician interaction in Korea in introduction (page 2, paragraph 3, lines 68–79).

“Triage staff members are organised according to the medical institution; in most Korean hospitals, nurses are key staff members who either work independently or with a physician [21]. Nurses can exchange opinions with other members of the triage staff, such as other nurses or physicians, during the triage process. This could help reduce triage errors and ensure decisions are made quickly [21]. For triage nurses, teamwork and collaboration with other healthcare professionals is an important clinical ability to enable emergency patients to receive suitable treatment [9,23]. Nurses and physicians are the primary providers involved in patient treatment, and effective cooperation between these professionals can ensure that nurses clearly and quickly interpret the condition of undiagnosed and unpredictable patients, both following physician instructions and effectively presenting their own opinions [9].”

Also, we re-analyzed the data about personnel involved in triage for clear understanding on triage staff (page 4–5, Table 1, lines 182–183).

Table 1. Differences in triage competency according to participant’s characteristics (N=152)

Comment 3: Method: I have doubts on the overall data analysis of the study, since I believe the simple regression model proposed may not be the best choice; if not already done, I suggest the Authors may check with a statistician.

Response 3: Thank you for your thorough suggestion. We checked with a statistician and reanalyzed the data using the multi-level (hierarchical) regression method (page 6, paragraph 1, lines 205–214).

“3.4. Factors affecting Triage Competency

Multiple regression analysis using the hierarchical method was conducted to identify the factors influencing triage competency among triage nurses. Critical thinking disposition, clinical judgement, and nurse–physician collaboration were entered as continuous independent variables. Categorical variables, including age, education level, experience in the emergency department, and triage experience, which showed significant differences in triage competency, were dummy-coded. Tolerance was above the cut-off value of 0.1 (range 0.44–0.69), the variance inflation factor was below the cut-off value of 10 (range 1.45–2.25), and the Durbin–Watson coefficient (1.948) was close to 2. This confirmed the absence of multicollinearity among the independent variables.”

Comment 4: The Authors mention that the nurses in the sample were selected among seven general hospitals. This creates a nested data structure (nurses clustered within hospitals). It means that observations within the same hospital are expected to be more similar to each other than observations from different hospitals. Ignoring such clustering might lead to underestimated standard errors and inflated significance levels. A multi-level model would be more appropriate to account for the non-independence of observations. Since one hospital may have contributed to many more participants than another, either in absolute numbers or in proportion, the data should be weighted to account for this imbalance.

Response 4: Thank you for your meaningful comments. Based on your suggestion, we analyzed the data using a hierarchical regression method. We entered the general characteristics, which showed significant differences in triage competency, as control variables in Model 1. Model 2 was created by entered independent variables. The manuscript was then revised according to the results (page 6, paragraphs 1-2, table 4, lines 205–230).

“3.4. Factors affecting Triage Competency

Multiple regression analysis using the hierarchical method was conducted to identify the factors influencing triage competency among triage nurses. Critical thinking disposition, clinical judgement, and nurse–physician collaboration were entered as continuous independent variables. Categorical variables, including age, education level, experience in the emergency department, and triage experience, which showed significant differences in triage competency, were dummy-coded. Tolerance was above the cut-off value of 0.1 (range 0.44–0.69), the variance inflation factor was below the cut-off value of 10 (range 1.45–2.25), and the Durbin–Watson coefficient (1.948) was close to 2. This confirmed the absence of multicollinearity among the independent variables.

In Model 1, general characteristics were used as control variables. These variables explained 25.5% (F=8.40, p<0.001) of the variance, and education level (≥ Master) (β=0.32, p=0.001) and experience in emergency department (≥6 years) (β=0.34, p=0.001) were influencing factors on triage competency. Model 2 was the result of the hierarchical entry of critical thinking disposition, clinical judgement, and nurse–physician collaboration. The factors influencing triage competency were education level (≥Master) (β=0.21, p=0.003), experience in emergency department (≥6 years) (β=0.32, p<0.001), critical thinking disposition (β=0.27, p<0.001), clinical judgement (β=0.16, p=0.008), and nurse–physician collaboration (β=-0.34, p<0.001). These variables explained 63.7% of the variance in triage competency among the triage nurses (F=27.52, p<0.001)

Table 4. Factors affecting triage competency (N=152)

Notes. B: unstandardised regression coefficient; ED: emergency department; SE: standard error; β: standardised regression coefficient; VIF: variance inflation factor; *Dummy variable: age (ref. = <30years), education level (ref. = associate degree), experience in the ED (ref. = <3years), triage experience (ref. = <12months)”

Comment 5: Result: As per previous points, Cluster-level information should also be provided. Please consider that hospital-level characteristics (e.g., size, resource availability) may influence both the predictors (e.g., critical thinking) and outcomes (triage competency).

Response 5: Thank you for your meaningful comments. Our manuscript studied triage nurses; each hospital has a limited number of nurses in this role. We therefore conducted the survey at multiple institutions in Chungcheong-so, South Korea, to ensure a powerful sample size. In conducting triage research, we considered not the size of the hospital but the type of emergency department system; therefore, we surveyed emergency medical centres. To consider the difference between the types of emergency center, we performed a t-test on dependent variables in Table 1. We found no differences in triage competency according to the type of emergency department. Based on your kind suggestions, we added hospital-level characteristics in 2.2 participants and setting section as tertiary and general hospitals in data collection (page 2, paragraph 6, lines 92-94).

“The participants in this study were triage nurses who had been working and performing triage for at least one year [25] in the emergency department of one tertiary hospital and six general hospitals in Chungcheong-do, South Korea.”

Comment 6: Discussion: The information on incentives for participation (A small token of gratitude was provided to the participants who completed the questionnaire) increases transparency. However, I believe Authors should evaluate the possibility for this to introduce selection bias (maybe those who respond to incentive are the most/least collaborating?). If this is a possibility, it should be at least mentioned in discussion. In alternative, the Authors could analyze non-respondents characteristics, or describe how the incentive was offered to mitigate bias (e.g., equal opportunity for all eligible participants).

Response 6: Thank you for your careful comments. Based on your advice, we describe the incentive for participation in the Limitation subsection. The token was a toothbrush set worth 3.43 USD (KRW 5,000 won), so it might not have created a bias (page 8, paragraph 3, lines 308–310).

“As a reward for participation, we gave each participant a toothbrush set worth 3.43 USD. While this incentive is small, it could have affected participation selection bias.”

END OF RESPONSES TO COMMENTS

Please let me know if you have any other concerns or questions about it. Thank you.

Reviewer 2 Report

Comments and Suggestions for Authors

COMPREHENSIVE MANUSCRIPT EVALUATION REPORT

I. General Considerations

Strengths:

  1. The study addresses a pertinent and timely topic in emergency nursing.
  2. Objectives are clearly delineated and align with the title and research problem.
  3. The overall methodology is appropriate for achieving the proposed objectives.
  4. Results are presented with clarity and objectivity.
  5. The discussion effectively correlates findings with extant literature.
  6. Conclusions are substantiated by the obtained results.

Weaknesses:

  1. Certain sections of the manuscript require further elaboration or refinement.
  2. Study limitations are not explicitly delineated.
  3. The currency of references could be enhanced.

General Recommendations for Improvement:

  1. Conduct a comprehensive review to ensure consistency and depth across all sections.
  2. Incorporate a dedicated section on study limitations.
  3. Update and diversify references, prioritizing publications from the last quinquennium.
  4. Consider including a concise systematic literature review to bolster the theoretical foundation.

II. Section-Specific Considerations

  1. Introduction and Objectives

Strengths:

  • Clearly defined objectives congruent with the title and investigated problem.
  • Adequate contextualization of the topic.

Recommendations for Enhancement:

- Consider incorporating a brief mention of the study's specific context (e.g., "...among triage nurses in emergency departments") in the objective statement.

2. Methodology

Strengths:

  • Study design appropriate to objectives.
  • Clear description of participants and inclusion criteria.
  • Presentation of original scale titles and reference sources.

Weaknesses:

  • Limited description of data collection procedures.
  • Insufficient detail on data analysis.
  • Limited information on psychometric properties of instruments.

Recommendations for Enhancement:

  • Elaborate on data collection procedures, including participant approach and collection environment.
  • Expand the description of data analysis, justifying the selection of statistical tests.
  • Include information on instrument validity, in addition to reliability.
  • Mention adaptation or validation processes for instruments in the specific context, if applicable.
  • Provide a brief description of each scale's structure (number of items, subscales, score interpretation).
  • Mention if permission was obtained for instrument use, where necessary.
  • Expand the ethical approval section, including details on informed consent and confidentiality protection.
  • Include a brief discussion on potential methodological limitations and how they were addressed.
  • Add a justification for sample size, possibly with a statistical power calculation.
  1. Results

Strengths:

  • Clear and objective presentation of key findings.
  • Effective use of tables for numerical data presentation.
  • Alignment with study objectives.

Recommendations for Enhancement:

  • Add a brief introductory paragraph in the results section, providing an overview of key findings.
  • Include more descriptive subheadings for each results subsection.
  • Consider including a figure or graph to visually illustrate some of the most important relationships.
  • Add a brief summary of the most significant findings at the end of the section.
  1. Discussion

Strengths:

  • Appropriate interpretation of findings in relation to objectives and existing literature.
  • Effective comparison with previous studies.
  • Address practical implications of results.

Recommendations for Enhancement:

  • Expand the discussion on implications of results for triage nurse education and training.
  • More explicitly address study limitations and how they may have influenced results.
  • Discuss potential future research directions based on findings.
  • Include a brief discussion on the applicability of results to different cultural contexts or healthcare systems.
  • Reflect on how findings may contribute to the development of policies or guidelines to improve triage practice.
  1. Conclusions

Strengths:

  • Clear presentation aligned with study objectives.
  • Effective synthesis of key findings.
  • Emphasis on practical implications.

Recommendations for Enhancement:

  • Include a brief statement on study limitations and their impact on result generalizability.
  • Propose specific directions for future research.
  • Elaborate further on implications for health policies and management practices in emergency departments.
  • Add a final reflection on the contribution of results to improving quality of care in emergencies.
  1. References

Strengths:

  • References pertinent to the investigated topic.
  • Inclusion of recent studies (last five years).

Weaknesses:

  • Proportion of recent references could be improved.
  • Possible lack of diversity in publication types.

Recommendations for Enhancement:

  • Increase the proportion of studies from the last five years.
  • Replace some older references with more recent versions, where possible.
  • Include a greater variety of publication types (original articles, systematic reviews, meta-analyses).
  • Add references from guidelines or reports of relevant organizations in emergency nursing.
  • Include references addressing emerging trends or current challenges in emergency triage.
  • Conduct a brief systematic literature review to ensure inclusion of all relevant and current references.
  1. Ethical Aspects

Strengths:

  • Clear mention of Research Ethics Committee approval.
  • Inclusion of protocol number and approval date.
  • Reference to the Declaration of Helsinki.

Recommendations for Enhancement:

  • Detail the process of obtaining informed consent.
  • Mention specific ethical considerations relevant to this type of study.
  • Include information on storage and protection of collected data.
  • Mention if there was compensation for study participation, if applicable.
  1. Contributions and Limitations

Weaknesses:

  • Lack of a clear section on study limitations.
  • Study contributions could be more explicitly highlighted.

Recommendations for Enhancement:

  • Add a specific section titled "Study Limitations" before the conclusion.
  • Discuss the cross-sectional nature of the study and its implications.
  • Mention possible geographical or cultural limitations of the sample.
  • Address potential biases, such as self-report bias in measures used.
  • Discuss limitations related to sample size or participant selection.
  • Expand the study contributions section, highlighting how results advance existing knowledge.
  • Elaborate on practical implications for nurse training and education.
  • Discuss how findings may influence policies and practices in emergency departments.
  • Use identified limitations to suggest specific directions for future studies.
  • Add a final paragraph synthesizing main contributions, acknowledging limitations, and reaffirming the study's importance.
  1. Data Analysis

Strengths:

  • Appropriate alignment with objectives and theoretical framework.
  • Proper use of statistical methods to investigate relationships between variables.
  • Comprehensive approach, including descriptive and inferential statistics.

Recommendations for Enhancement:

  • Consider including mediation or moderation analysis to explore possible indirect effects between variables.
  • Evaluate the possibility of conducting subgroup analysis to identify if relationships between variables differ based on specific participant characteristics.
  • Include a brief justification for the choice of specific statistical methods used, directly relating them to study objectives.
  1. Comparison of Results, Data, and Literature

Strengths:

  • Effective integration of results and existing literature.
  • Specific comparisons with previous studies.
  • Evidence-based interpretation.
  • Critical approach when results differ or add new perspectives.

Recommendations for Enhancement:

  • Consider including a brief meta-analysis or recent systematic review on the topic.
  • Explore more deeply the theoretical implications of results, possibly proposing adjustments or extensions to existing conceptual models.
  • Discuss how findings align with or challenge current trends in emergency nursing practice and research.
  • Explicitly address how results contribute to filling specific gaps in the literature identified in the introduction.
  1. Manuscript Structure and Organization

Strengths:

  • Adequate general structure, following standard format of scientific articles.
  • Well-defined sections and logical presentation sequence.

Recommendations for Enhancement:

  • Consider adding more descriptive subheadings in each main section to improve navigability.
  • Ensure smooth transition between sections, possibly adding linking phrases.
  • Review consistency in terminology use throughout the manuscript.
  • Consider including a flowchart or diagram to illustrate participant selection process or study flow.
  1. Relevance and Originality

Strengths:

  • Addressing a relevant topic for emergency nursing practice.
  • Investigation of multiple factors influencing triage competency.

Recommendations for Enhancement:

  • -More explicitly highlight the study's originality in relation to existing literature.
  • Discuss how findings may contribute to the development of specific interventions to improve triage competency.
  • Consider including a brief section on implications of results for nursing education and continuous professional development.
  1. Practical Applicability

Strengths:

  • Discussion of practical implications of results.
  • Focus on improving triage competency in emergency settings.

Recommendations for Enhancement:

  • Elaborate more specific recommendations for clinical practice based on results.
  • Discuss how findings can be incorporated into training programs and continuing education for triage nurses.
  • Consider including a box or table summarizing main practical implications and recommendations.
  1. Clarity and Writing Style

Strengths:

  • Generally clear and objective writing.
  • Appropriate use of scientific terminology.

Recommendations for Enhancement:

 Review the manuscript to ensure consistency in use of technical terms.

  • Consider simplifying some complex sentences to improve readability.
  • Ensure all acronyms and abbreviations are defined at first occurrence.
  • Conduct a careful review to correct any grammatical or punctuation errors.
  1. Tables and Figures

Strengths:

- Effective use of tables to present numerical data.

Recommendations for Enhancement:

  • Consider adding figures or graphs to visually illustrate the most important relationships found in the study.
  • Ensure all tables are self-explanatory, with clear titles and footnotes when necessary.
  • Check formatting and style of tables to ensure consistency with journal guidelines.
  1. Cultural and Contextual Considerations

Recommendations for Enhancement:

  • Discuss how results may be applicable or vary in different cultural contexts or healthcare systems.
  • Consider including a brief discussion on how cultural factors may influence triage competency and interprofessional collaboration.
  • Address possible implications of results for resource-limited contexts or developing healthcare systems.
  1. Future Perspectives

Recommendations for Enhancement:

  • Elaborate more detailed directions for future research based on study findings.
  • Discuss potential longitudinal studies that could expand current results.
  • Consider suggesting interventional studies based on factors identified as influential in triage competency.
  1. Additional Ethical Aspects

Recommendations for Enhancement:

  • Discuss any specific ethical considerations related to data collection in emergency settings.
  • Address how the study ensured that nurses' participation did not negatively affect their work responsibilities.
  • Mention if there was any procedure to offer feedback or study results to participants.

Implementation of these additional suggestions will contribute to a more robust, comprehensive, and impactful manuscript, strengthening its contribution to the field of emergency nursing and triage competency.

Author Response

Response to Reviewer 2 Comments

My co-author and I wish to re-submit our revised manuscript entitled “Effects of critical thinking disposition, clinical judgement, and nurse–physician collaboration on triage competency among triage nurses” with changes that thoroughly address your comments.

We thank you for your thoughtful suggestions and insights. The manuscript has benefited from your thorough feedback. We look forward to working with you to move this manuscript closer to publication in Healthcare.

The manuscript has been rechecked and the necessary changes have been made in accordance with your suggestions. The responses to all comments have been prepared and attached herewith/given below.

We appreciate your consideration.

Comment 1: General Considerations-Conduct a comprehensive review to ensure consistency and depth across all sections.

Response 1: Thank you for your detailed comments. Based on your recommendations, we have thoroughly reviewed and revised the entire manuscript.

Comment 2: General Considerations-Incorporate a dedicated section on study limitations.

Response 2: Thank you for your kind recommendation. We added a Study Limitations subsection before the Conclusion section (page 8, paragraph 3, lines 301–312).

4.1. Study Limitations

Since this study only included triage nurses working in emergency departments in a certain region, the findings are limited in terms of their generalisability. Therefore, they should be validated with broader or different cultural populations across more diverse regions. In addition, we collected cross-sectional data using a self-reported survey; a long-term observation study should be considered. Moreover, we could not consider institutional characteristics, so the size and available resources of different emergency departments might be considered in future research. As a reward for participation, we gave each participant a toothbrush set worth 3.43 USD. While this incentive is small, it could have affected participation selection bias. Finally, as we recruited participants who worked in emergency departments through convenience sampling, randomisation should be considered to check for internal validity.“

Comment 3: General Considerations-Update and diversify references, prioritizing publications from the last quinquennium.

Response 3: Thank you for your kind recommendation. We have attempted to use references from the last five years. We ask for your understanding for the following references, which are older than that. Four references [27,30–32] relate to original instruments, and three [8,12,15] are important references for the concept of critical thinking disposition. References 17 (2019), 19 (2018), and 35(2018) were published within the last seven years, but are needed to describe these manuscripts. (pages 9-10, references 1-38, lines 343–421).

Comment 4: General Considerations-Consider including a concise systematic literature review to bolster the theoretical foundation.

Response 4: Thank you for your kind recommendation. We reviewed various literature, including systematic literature [4] and quantitative integrative review [3,11,13, 38] (pages 9-10, references 1-38, lines 343–421).

  1. Reblora, J.M.; Lopez, V.; Goh, Y.-S. Experiences of nurses working in a triage area: an integrative review.  Crit. Care202033, 567–575, doi:10.1016/j.aucc.2020.01.005.
  2. Fekonja, Z.; Kmetec, S.; Fekonja, U.; Mlinar Reljić, N.; Pajnkihar, M.; Strnad, M. Factors contributing to patient safety during triage process in the emergency department: a systematic review.  Clin. Nurs.202332, 5461–5477, doi:10.1111/jocn.16622.
  3. Gorick, H. Factors that affect nurses’ triage decisions in the emergency department: a literature review.  Nurse202230, 14–19, doi:10.7748/en.2022.e2123.
  4. Lee, Y.; Oh, Y. Levels, antecedents, and consequences of critical thinking among clinical nurses: a quantitative literature review.  Educ. Eval. Health Prof.202017, 26, doi:10.3352/jeehp.2020.17.26.
  5. López Hernández, M.; Puig-Llobet, M.; Higon Fernández, S.; Franco Freirut, M.; Moreno Mateos, Y.; Galimany Masclans, Patient satisfaction with the level of competence of the triage nurse in hospital emergency departments. J. Clin. Nurs. 2024, Published online ahead of print. doi: 10.1111/jocn.17605.

Comment 5: Introduction and Objectives-Consider incorporating a brief mention of the study's specific context (e.g., "...among triage nurses in emergency departments") in the objective statement.

Response 5: According to your kind suggestion, we added the specific context of the study in an objective statement (page 2, paragraph 6, lines 83–86).

“This study aimed to investigate the levels of critical thinking disposition, clinical judgment, nurse–physician collaboration, and triage competency in triage nurses working in the emergency department and to determine the factors affecting triage competency among these nurses.”

Comment 6: Methodology-Elaborate on data collection procedures, including participant approach and collection environment.

Response 6: Thank you for your suggestions. We have therefore added data collection procedures, specific setting, and approach to the manuscript (page 2, paragraph 8, lines 91-94), (pages 3-4, paragraphs 7-1, lines 142-153), (page 4, paragraph 3, 161-173).

“2.2. Participants and Setting

The participants in this study were triage nurses who had been working and performing triage for at least one year [25] in the emergency department of one tertiary hospital and six general hospitals in Chungcheong-do, South Korea.”

“2.4. Data Collection

The data for this study were collected between 1 September and 1 November 2023, after receiving approval from an institutional review board. We first obtained permission for data collection from the nursing and emergency department heads of the participating hospitals. The first author reached out to each emergency department and explained the study objectives and methods with written information. Triage nurses who voluntarily consented to participate signed informed consent forms and were provided with a questionnaire for data collection. To protect the privacy of the participants, data collection was conducted in a separate space, and the responses to the questionnaire were received in a sealed envelope; furthermore, an identification number was assigned in place of the participant’s name. A small token of gratitude (i.e. a toothbrush set worth 5000 KRW [3.43 USD]) was provided to the participants who completed the questionnaire.”

“2.6. Ethical Considerations

This study was reviewed and approved by an institutional review board at the researchers’ affiliated institution (2023-06-021-001). The participants received a written explanation of the study objectives and methods, and they voluntarily signed two written informed consent form. One signed informed consent form was then provided to the participant. The explanation stated that participation in the study would involve no expected risks, that participants could withdraw at any time during the study, and that withdrawal would not result in any disadvantages. The participants were informed that the questionnaires would be assigned arbitrary identification numbers for management, would only be used for research purposes, and would be stored in a locked cabinet for data protection. In addition, they were informed that the privacy of their personal responses and information would be thoroughly maintained. Authors collected the data either before or after each participant’s shift to avoid negatively affecting their work responsibilities.”

Comment 7: Methodology-        Expand the description of data analysis, justifying the selection of statistical tests.

Response 7: Thank you for your comments. Based on your comment and reviewer 1 suggestion, we analyzed the data using a hierarchical regression method to control the sampling imbalance. We entered the general characteristics, which showed significant differences in triage competency, as control variables in Model 1. Model 2 was created by entered independent variables. The manuscript was then revised according to the results (page 6, paragraphs 1-2, table 4, lines 205–230).

“3.4. Factors affecting Triage Competency

Multiple regression analysis using the hierarchical method was conducted to identify the factors influencing triage competency among triage nurses. Critical thinking disposition, clinical judgement, and nurse–physician collaboration were entered as continuous independent variables. Categorical variables, including age, education level, experience in the emergency department, and triage experience, which showed significant differences in triage competency, were dummy-coded. Tolerance was above the cut-off value of 0.1 (range 0.44–0.69), the variance inflation factor was below the cut-off value of 10 (range 1.45–2.25), and the Durbin–Watson coefficient (1.948) was close to 2. This confirmed the absence of multicollinearity among the independent variables.

In Model 1, general characteristics were used as control variables. These variables explained 25.5% (F=8.40, p<0.001) of the variance, and education level (≥ Master) (β=0.32, p=0.001) and experience in emergency department (≥6 years) (β=0.34, p=0.001) were influencing factors on triage competency. Model 2 was the result of the hierarchical entry of critical thinking disposition, clinical judgement, and nurse–physician collaboration. The factors influencing triage competency were education level (≥Master) (β=0.21, p=0.003), experience in emergency department (≥6 years) (β=0.32, p<0.001), critical thinking disposition (β=0.27, p<0.001), clinical judgement (β=0.16, p=0.008), and nurse–physician collaboration (β=-0.34, p<0.001). These variables explained 63.7% of the variance in triage competency among the triage nurses (F=27.52, p<0.001)

Table 4. Factors affecting triage competency (N=152)

Notes. B: unstandardised regression coefficient; ED: emergency department; SE: standard error; β: standardised regression coefficient; VIF: variance inflation factor; *Dummy variable: age (ref. = <30years), education level (ref. = associate degree), experience in the ED (ref. = <3years), triage experience (ref. = <12months)”

Comment 8: Methodology-        Include information on instrument validity, in addition to reliability.

Response 8: We added information about validity for the instruments. These instruments were validated when the scale was developed (page 3 paragraphs 3-6, lines 109-111, 117-119, 125-128, 134-136).

“2.3.2 Critical Thinking Disposition

Critical thinking disposition was measured using the critical thinking disposition instrument developed and validated by Yoon [27].

2.3.3. Clinical Judgement

Clinical judgement was measured using the clinical judgement scale developed and validated by Kwon and Park [29].

2.3.4. Nurse–Physician Collaboration

Nurse–physician collaboration was measured using the Korean Version of the Nurse–Physician Collaboration Scale (K-NPCS), which was translated and validated by Mun [30] from the Nurse–Physician Collaboration Scale (NPCS) developed by Ushiro [31].

2.3.5. Triage Competency

Triage competency was measured using the triage competency scale for emergency department nurses developed and validated by Moon and Park [32].”

Comment 9: Methodology-Mention adaptation or validation processes for instruments in the specific context, if applicable.

Response 9: We are grateful for your thorough review. We have added Cronbach’s α value for triage nurses or emergency nurses developed (page 3 paragraphs 3-6, lines 109, 114-117, 123-125, 131-134, 140-141).

“2.3.2 Critical Thinking Disposition

In terms of the reliability of the instrument, Cronbach’s α was 0.84 at the time of development [27], 0.90 for triage nurses [28], and 0.84 in this study.

2.3.3. Clinical Judgement

Cronbach’s α was 0.91 at the time of development for nurses [29], and 0.92 in this study.

2.3.4. Nurse–Physician Collaboration

In terms of reliability, Cronbach’s α was 0.8 for the original instrument in the study by Ushiro [31], 0.95 in the study by Mun [30], 0.92 for emergency nurse [9], and 0.96 in this study.

2.3.5. Triage Competency

In terms of reliability, Cronbach’s α was 0.91 at the time of development [32], 0.91 for triage nurses [28] and 0.97 in this study.”

Comment 10: Methodology-Provide a brief description of each scale's structure (number of items, subscales, score interpretation).

Response 10: Thank you for your kind recommendation. We have added descriptions of the structure, including number of items, subscales, and score interpretation, for each scale (page 3 paragraphs 3-6, lines 109, 111-114, 117, 119-123, 125, 129-131, 134, 136-139).

“2.3.2 Critical Thinking Disposition

This instrument consists of 27 items, including intellectual passion and curiosity, prudence, healthy scepticism, self-confidence, intellectual fairness, objectivity, and systematicity. Each item is scored on a 5-point Likert scale, with higher scores indicating a stronger critical thinking disposition.

2.3.3. Clinical Judgement

This scale consists of 23 items, including integrated data analysis, intervention evaluation and reflections, intervention evidence, inter-professional collaboration, patient-centred nursing, and collaboration between nursing colleagues. Each item is scored on a 5-point Likert scale, with higher scores indicating better clinical judgement ability.

2.3.4. Nurse–Physician Collaboration

This scale consists of 27 items including information sharing, decision-making about treatment and care, and nurse–physician relationships. Each item is scored on a 5-point Likert scale, with lower scores indicating better collaboration.

2.3.5. Triage Competency

This scale consists of 30 items including clinical judgement, communication, professional identification, personal coping, and management. Each item is scored on a 5-point Likert scale, with possible scores ranging from 30 to 150 points, where higher scores indicate better triage competency.

Comment 11: Methodology- Mention if permission was obtained for instrument use, where necessary.

Response 11: Based on your suggestion, we added a description in the measures section (page 3, paragraph 2, line 103).

2.3 Measures

All instruments were used after obtaining permission from their authors.”

Comment 12: Methodology- Expand the ethical approval section, including details on informed consent and confidentiality protection.

Response 12: We revised informed consent and confidentiality protection in the Ethical Approval subsection (page 4, paragraph 3, lines 161–173).

“2.6. Ethical Considerations

This study was reviewed and approved by an institutional review board at the researchers’ affiliated institution (2023-06-021-001). The participants received a written explanation of the study objectives and methods, and they voluntarily signed two written informed consent form. One signed informed consent form was then provided to the participant. The explanation stated that participation in the study would involve no expected risks, that participants could withdraw at any time during the study, and that withdrawal would not result in any disadvantages. The participants were informed that the questionnaires would be assigned arbitrary identification numbers for management, would only be used for research purposes, and would be stored in a locked cabinet for data protection. In addition, they were informed that the privacy of their personal responses and information would be thoroughly maintained. Authors collected the data either before or after each participant’s shift to avoid negatively affecting their work responsibilities.”

Comment 13: Methodology- Include a brief discussion on potential methodological limitations and how they were addressed.

Response 13: According to your suggestion, we have added methodological limitations, including generalizability, in the Study Limitations subsection (page 8, paragraph 3, lines 301–312).

4.1. Study Limitations

Since this study only included triage nurses working in emergency departments in a certain region, the findings are limited in terms of their generalisability. Therefore, they should be validated with broader or different cultural populations across more diverse regions. In addition, we collected cross-sectional data using a self-reported survey; a long-term observation study should be considered. Moreover, we could not consider institutional characteristics, so the size and available resources of different emergency departments might be considered in future research. As a reward for participation, we gave each participant a toothbrush set worth 3.43 USD. While this incentive is small, it could have affected participation selection bias. Finally, as we recruited participants who worked in emergency departments through convenience sampling, randomisation should be considered to check for internal validity.”

Comment 14: Methodology- Add a justification for sample size, possibly with a statistical power calculation.

Response 14: We described a justification for sample size. We used Cohen’s moderate effect size (pages 2-3, paragraphs 7-1, lines 95–101).

“To calculate the required sample size, we used G*power 3.1.9.7 for multiple regression analysis, assuming a power of .90, a significance level of .05, a Cohen’s [26] moderate effect size of .15, and 11 predictive variables (eight general characteristics and three independent variables). Consequently, the required sample size was found to be 152 individuals. To account for a dropout rate of 10%, we distributed questionnaires to 167 individuals. We received 162 responses (97%), and after excluding 10 insincere responses, we included 152 individuals in the final analysis.”

Comment 15: Results- Add a brief introductory paragraph in the results section, providing an overview of key findings.

Response 15: Thank you for your thoughtful advice. We have included a summary of the results in the beginning of the Discussion section. We did not want to add a similar paragraph in the Results section, as it would create duplication and a less concise manuscript. We appreciate your understanding on this point (page 7, paragraph 2, lines 232-236).

“4. Discussion

This study aimed to investigate the factors affecting triage competency in triage nurses. Consequently, nurse–physician collaboration, experience in the emergency department, critical thinking disposition, education level, and clinical judgement were found to affect triage competency in descending order. The regression analysis showed that these factors explained 63.7% of the variance in triage competency.”

Comment 16: Results- Include more descriptive subheadings for each results subsection.

Response 16: According to your suggestion, we have revised the subheadings in the results section to be more descriptive (pages 4-6, lines 174-175, 185-186, 196-197, 205).

“3. Results

3.1. Differences in Triage Competency According to Participants’ Characteristics

3.2. Descriptive Statistics of Critical Thinking Disposition, Clinical Judgment, and Nurse–Physician Collaboration on Triage Competency

3.3. Correlations among Critical Thinking Disposition, Clinical Judgment, and Nurse–Physician Collaboration with Triage Competency

3.4. Factors affecting Triage Competency”

Comment 17: Results-Consider including a figure or graph to visually illustrate some of the most important relationships.

Response 17: Thank you for your kind recommendation. It is difficult to illustrate the main results of hierarchical regression, so we have chosen to present the information as a Table. We appreciate your understanding on this point.

Comment 18: Results-Add a brief summary of the most significant findings at the end of the section.

Response 18: Thank you for your thoughtful advice. We have provided a summary of the results in the beginning of the Discussion section. We would prefer not to add a summary to the end of the Results section, as it would create duplication. We appreciate your understanding on this point (page 7, paragraph 2, lines 232-236).

“4. Discussion

This study aimed to investigate the factors affecting triage competency in triage nurses. Consequently, nurse–physician collaboration, experience in the emergency department, critical thinking disposition, education level, and clinical judgement were found to affect triage competency in descending order. The regression analysis showed that these factors explained 63.7% of the variance in triage competency.”

Comment 19: Discussion-Expand the discussion on implications of results for triage nurse education and training.

Response 19: According to your suggestion, we expanded the discussion about education for triage nurses (page 7, paragraph 3, lines 267-271), (page 8, paragraph1, lines 280-285).

“In a previous study, emergency department triage nurses who had received web-based triage education showed better clinical judgement [6]. Clinical judgement is a core competency of nurses, enabling them to assess patients’ urgency and determine priorities for easy access intervention. Therefore, continuous education programs using web-based methods are an essential piece of effectively improving clinical judgement.”

“Korean triage nurses need systematic training management to improve their competency for triage [35]. Ongoing professional development education is necessary for enhancing their skills [38]. Furthermore, to ensure expertise based on clinical experience for triage nurses in the emergency department, transfer to other departments should be restricted, and continuous education should be provided to improve triage accuracy [6].”

Comment 20: Discussion-More explicitly address study limitations and how they may have influenced results.

Response 20: Thank you for your kind recommendation. We have separated the Study Limitations subsection and placed it before the Conclusion section. It has been more explicitly tied to the results (page 8, paragraph 3, lines 301–312).

4.1. Study Limitations

Since this study only included triage nurses working in emergency departments in a certain region, the findings are limited in terms of their generalisability. Therefore, they should be validated with broader or different cultural populations across more diverse regions. In addition, we collected cross-sectional data using a self-reported survey; a long-term observation study should be considered. Moreover, we could not consider institutional characteristics, so the size and available resources of different emergency departments might be considered in future research. As a reward for participation, we gave each participant a toothbrush set worth 3.43 USD. While this incentive is small, it could have affected participation selection bias. Finally, as we recruited participants who worked in emergency departments through convenience sampling, randomisation should be considered to check for internal validity.”

Comment 21: Discussion-Discuss potential future research directions based on findings.

Response 21: Based on your suggestion, we revised the Discussion section to include information about potential future research (page 7, paragraph 3, lines 250-253), (page 8, paragraph 1, lines 288-292).

“In this regard, given that mutual collaboration between physicians and nurses affects the work of nurses [33], promoting teamwork between healthcare professionals is crucial to improving triage competency [4] as a professional capability of triage nurses. The experience of nurse–physician collaboration in emergency department in future research.”

“Nurses who take a degree course to improve their expertise face new challenges, and in this process, they gain self-confidence and value their own competencies more highly [37]. Therefore, further research is needed to determine whether education level is a factor in improving triage competency so that active support and encouragement can be provided within nursing organisations.”

Comment 22: Discussion-Include a brief discussion on the applicability of results to different cultural contexts or healthcare systems.

Response 22: Based on your suggestion, we added the applicability in the Study Limitations subsection (page 8, paragraph 3, lines 301–305).

4.1. Study Limitations

Since this study only included triage nurses working in emergency departments in a certain region, the findings are limited in terms of their generalisability. Therefore, they should be validated with broader or different cultural populations across more diverse regions.”

Comment 23: Discussion-Reflect on how findings may contribute to the development of policies or guidelines to improve triage practice.

Response 23: Based on your suggestion, we revised the discussion about developing policies (page 8, paragraph 1, lines 290-292), (page 8, paragraph 2, lines 297-299).

“Therefore, further research is needed to determine whether education level is a factor in improving triage competency so that active support and encouragement can be provided within nursing organisations.”

“If validated and confirmed in future studies, our findings could help prepare institutions and establish systems to support nurse-physician collaboration, to provide ongoing training triage competency among triage nurses in emergency departments.”

Comment 24: Conclusions-Include a brief statement on study limitations and their impact on result generalizability.

Response 24: Thank you for your kind recommendation. Based on your Comment 2, we included a Study Limitations subsection before the Conclusion section, which mentions generalizability (page 8, paragraph 3, lines 301–302).

4.1. Study Limitations

Since this study only included triage nurses working in emergency departments in a certain region, the findings are limited in terms of their generalisability.”

Comment 25: Conclusions-Propose specific directions for future research.

Response 25: We revised the Conclusion section to propose specific directions for future research based on the results (page 8, paragraph 2, lines 296–300).

“Consequently, various factors were identified to direct influence triage competency. If validated and confirmed in future studies, our findings could help prepare institutions and establish systems to support nurse-physician collaboration, to provide ongoing training triage competency among triage nurses in emergency departments.”

Comment 26: Conclusions-Elaborate further on implications for health policies and management practices in emergency departments.

Response 26: We described implications for health policies in Korean emergency departments based on the results (page 8, paragraph 2, lines 297-300), (page 8, paragraph 4, lines 321-325).

“If validated and confirmed in future studies, our findings could help prepare institutions and establish systems to support nurse-physician collaboration, to provide ongoing training triage competency among triage nurses in emergency departments.”

“Therefore, organisational policies that promote the formation of collaborative relationships among healthcare workers in the triage area are of the utmost importance. Furthermore, various educational interventions should be provided to enhance critical thinking and clinical judgement, and continuing education opportunities should be available at the postgraduate level for triage nurses.”

Comment 27: Conclusions-Add a final reflection on the contribution of results to improving quality of care in emergencies.

Response 27: According to your suggestion, we added a description about quality of care in the emergency department (page 8, paragraph 4, lines 319–323).

“These factors contributed to nurses’ triage expertise, which influences patients’ safety and the quality of care received in emergency departments. Therefore, organisational policies that promote the formation of collaborative relationships among healthcare workers in the triage area are of the utmost importance.”

Comment 28: References-Increase the proportion of studies from the last five years.

Response 28: Thank you for your kind recommendation. We tried to use references from the last five years. We appreciate your understanding regarding the following references. Four [27,30–32] references relate to the original instruments, and three [8,12,15] are important references for the concept of critical thinking disposition. References 17 (2019), 19 (2018), and 35 (2018) were published within seven years, but they are needed to describe these manuscripts (pages 9-10, references 1-38, lines 343–421).

Comment 29: References-Replace some older references with more recent versions, where possible.

Response 29: Thank you for your kind recommendation. We tried to use references from the last five years. We appreciate your understanding regarding the following references. Four [27,30–32] references relate to the original instruments, and three [8,12,15] are important references for the concept of critical thinking disposition. References 17 (2019), 19 (2018), and 35 (2018) were published within seven years, but they are needed to describe these manuscripts.

Comment 30: References-Include a greater variety of publication types (original articles, systematic reviews, meta-analyses).

Response 30: We are grateful for your thorough review. Most of the references we used were original articles, and we referenced a systematic review [4], and quantitative integrative reviews [3,11,13,38] (pages 9-10, references 1-38, lines 343–421).

  1. Reblora, J.M.; Lopez, V.; Goh, Y.-S. Experiences of nurses working in a triage area: an integrative review.  Crit. Care202033, 567–575, doi:10.1016/j.aucc.2020.01.005.
  2. Fekonja, Z.; Kmetec, S.; Fekonja, U.; Mlinar Reljić, N.; Pajnkihar, M.; Strnad, M. Factors contributing to patient safety during triage process in the emergency department: a systematic review.  Clin. Nurs.202332, 5461–5477, doi:10.1111/jocn.16622.
  3. Gorick, H. Factors that affect nurses’ triage decisions in the emergency department: a literature review.  Nurse202230, 14–19, doi:10.7748/en.2022.e2123.
  4. Lee, Y.; Oh, Y. Levels, antecedents, and consequences of critical thinking among clinical nurses: a quantitative literature review.  Educ. Eval. Health Prof.202017, 26, doi:10.3352/jeehp.2020.17.26.
  5. López Hernández, M.; Puig-Llobet, M.; Higon Fernández, S.; Franco Freirut, M.; Moreno Mateos, Y.; Galimany Masclans, Patient satisfaction with the level of competence of the triage nurse in hospital emergency departments. J. Clin. Nurs. 2024, Published online ahead of print. doi: 10.1111/jocn.17605.

Comment 31: References-Add references from guidelines or reports of relevant organizations in emergency nursing.

Response 31: Thank you for your kind recommendation. We added a reference for the Korean Triage and Acuity Scale manual [21] (pages 9-10, references 1-38, lines 343–421).

  1. Korean Triage and Acuity Scale committee. Korean triage and acuity scale manual, 2nd; Koonja: Paju, Gyeonggi-do, Korea, 2021.

Comment 32: References-Include references addressing emerging trends or current challenges in emergency triage.

Response 32: We included the triage trend and current challenges in emergency departments for triage with the following references for Korea [1,6,9, 21] and other countries [2,4,11,38] (pages 9-10, references 1-38, lines 343–421).

  1. Chang, H.; Yu, J.Y.; Lee, G.H.; Heo, S.; Lee, S.U.; Hwang, S.Y.; Yoon, H.; Cha, W.C.; Shin, T.G.; Sim, M.S.; et al. Clinical Support system for triage based on federated learning for the Korea triage and acuity scale. Heliyon20239, e19210, doi:10.1016/j.heliyon.2023.e19210.
  2. Moon, S.-H.; Cho, I.-Y. The effect of competency-based triage education application on emergency nurses’ triage competency and performance. Healthcare (Basel)202210, 596, doi:10.3390/healthcare10040596.
  3. Hwang, S.; Shin, S. Factors affecting triage competence among emergency room nurses: a cross-sectional study.  Clin. Nurs.202332, 3589–3598, doi:10.1111/jocn.16441.
  4. Moon, S.H.; Jeon, M.K.; Ju, Facilitators and barriers of the triage process based on emergency nurses' experience with the Korean triage and acuity scale: A qualitative content analysis. Asian Nurs. Res. 2021, 15, 255-264, doi: 10.1016/j.anr.2021.08.001.

  1. Moura, B.R.S.; Oliveira, G.N.; Medeiros, G.; Vieira, A. de S.; Nogueira, L. de S. Rapid triage performed by nurses: signs and symptoms associated with identifying critically ill patients in the emergency department.  J. Nurs. Pract.202228, e13001, doi:10.1111/ijn.13001.
  2. Fekonja, Z.; Kmetec, S.; Fekonja, U.; Mlinar Reljić, N.; Pajnkihar, M.; Strnad, M. Factors contributing to patient safety during triage process in the emergency department: a systematic review.  Clin. Nurs.202332, 5461–5477, doi:10.1111/jocn.16622.
  3. Gorick, H. Factors that affect nurses’ triage decisions in the emergency department: a literature review.  Nurse202230, 14–19, doi:10.7748/en.2022.e2123.
  4. López Hernández, M.; Puig-Llobet, M.; Higon Fernández, S.; Franco Freirut, M.; Moreno Mateos, Y.; Galimany Masclans, Patient satisfaction with the level of competence of the triage nurse in hospital emergency departments. J. Clin. Nurs. 2024, Published online ahead of print. doi: 10.1111/jocn.17605.

Comment 33: References-Conduct a brief systematic literature review to ensure inclusion of all relevant and current references.

Response 33: We are grateful for your thorough review. Most of the references we used were original articles, and we referenced a systematic review [4], and quantitative integrative reviews [3,11,13,38] (pages 9-10, references 1-38, lines 343–421).

  1. Reblora, J.M.; Lopez, V.; Goh, Y.-S. Experiences of nurses working in a triage area: an integrative review.  Crit. Care202033, 567–575, doi:10.1016/j.aucc.2020.01.005.
  2. Fekonja, Z.; Kmetec, S.; Fekonja, U.; Mlinar Reljić, N.; Pajnkihar, M.; Strnad, M. Factors contributing to patient safety during triage process in the emergency department: a systematic review.  Clin. Nurs.202332, 5461–5477, doi:10.1111/jocn.16622.
  3. Gorick, H. Factors that affect nurses’ triage decisions in the emergency department: a literature review.  Nurse202230, 14–19, doi:10.7748/en.2022.e2123.
  4. Lee, Y.; Oh, Y. Levels, antecedents, and consequences of critical thinking among clinical nurses: a quantitative literature review.  Educ. Eval. Health Prof.202017, 26, doi:10.3352/jeehp.2020.17.26.
  5. López Hernández, M.; Puig-Llobet, M.; Higon Fernández, S.; Franco Freirut, M.; Moreno Mateos, Y.; Galimany Masclans, Patient satisfaction with the level of competence of the triage nurse in hospital emergency departments. J. Clin. Nurs. 2024, Published online ahead of print. doi: 10.1111/jocn.17605.

Comment 34: Ethical Aspects-Detail the process of obtaining informed consent.

Response 34: We have provided a detailed discussion of the informed consent procedure based on Comment 12 (page 4, paragraph 3, lines 161–173).

“2.6. Ethical Considerations

This study was reviewed and approved by an institutional review board at the researchers’ affiliated institution (2023-06-021-001). The participants received a written explanation of the study objectives and methods, and they voluntarily signed two written informed consent form. One signed informed consent form was then provided to the participant. The explanation stated that participation in the study would involve no expected risks, that participants could withdraw at any time during the study, and that withdrawal would not result in any disadvantages. The participants were informed that the questionnaires would be assigned arbitrary identification numbers for management, would only be used for research purposes, and would be stored in a locked cabinet for data protection. In addition, they were informed that the privacy of their personal responses and information would be thoroughly maintained. Authors collected the data either before or after each participant’s shift to avoid negatively affecting their work responsibilities.”

Comment 35: Ethical Aspects-Mention specific ethical considerations relevant to this type of study.

Response 35: We described confidentiality protection in the ethical approval subsection based on the descriptive cross-sectional study type such as expected risks, withdraw, disadvantages, security, etc. (page 4, paragraph 3, lines 166–173).

“The explanation stated that participation in the study would involve no expected risks, that participants could withdraw at any time during the study, and that withdrawal would not result in any disadvantages. The participants were informed that the questionnaires would be assigned arbitrary identification numbers for management, would only be used for research purposes, and would be stored in a locked cabinet for data protection. In addition, they were informed that the privacy of their personal responses and information would be thoroughly maintained. Authors collected the data either before or after each participant’s shift to avoid negatively affecting their work responsibilities.”

Comment 36: Ethical Aspects-Include information on storage and protection of collected data.

Response 36: Thank you for your suggestion. We added information on storage for data protection (page 4, paragraph 3, lines 168–170).

“The participants were informed that the questionnaires would be assigned arbitrary identification numbers for management, would only be used for research purposes, and would be stored in a locked cabinet for data protection.”

Comment 37: Ethical Aspects-Mention if there was compensation for study participation, if applicable.

Response 37: We are grateful for your thorough review. We described the compensation for participation in data collection and added a related bias in the Study Limitation subsection based on Reviewer 1’s comment (page 4, paragraph 1, lines 152–153), (page 8, paragraph 3, 308–310).

“A small token of gratitude (i.e. a toothbrush set worth 5000 KRW [3.43 USD]) was provided to the participants who completed the questionnaire.”

“As a reward for participation, we gave each participant a toothbrush set worth 3.43 USD. While this incentive is small, it could have affected participation selection bias.”

Comment 38: Contributions and Limitations-Add a specific section titled "Study Limitations" before the conclusion.

Response 38: Thank you for your kind recommendation. We added a Study Limitation subsection before the Conclusions section based on Comment 2 (page 8, paragraph 3, lines 301-312)

4.1. Study Limitations

Since this study only included triage nurses working in emergency departments in a certain region, the findings are limited in terms of their generalisability. Therefore, they should be validated with broader or different cultural populations across more diverse regions. In addition, we collected cross-sectional data using a self-reported survey; a long-term observation study should be considered. Moreover, we could not consider institutional characteristics, so the size and available resources of different emergency departments might be considered in future research. As a reward for participation, we gave each participant a toothbrush set worth 3.43 USD. While this incentive is small, it could have affected participation selection bias. Finally, as we recruited participants who worked in emergency departments through convenience sampling, randomisation should be considered to check for internal validity.”

Comment 39: Contributions and Limitations-Discuss the cross-sectional nature of the study and its implications.

Response 39: Based on your advice, we added the cross-sectional nature of the study in the Limitation subsection (page 8, paragraph 3, lines 305–306).

“In addition, we collected cross-sectional data using a self-reported survey; a long-term observation study should be considered.”

Comment 40: Contributions and Limitations-Mention possible geographical or cultural limitations of the sample.

Response 40: Based on your advice, we added cultural limitations (page 8, paragraph 3, lines 302-305).

“Since this study only included triage nurses working in emergency departments in a certain region, the findings are limited in terms of their generalisability. Therefore, they should be validated with broader or different cultural populations across more diverse regions.”

Comment 41: Contributions and Limitations-Address potential biases, such as self-report bias in measures used.

Response 41: According to your suggestion, we added potential biases, including self-report and participation reward (page 8, paragraph 3, lines 305–306).

“In addition, we collected cross-sectional data using a self-reported survey; a long-term observation study should be considered.”

Comment 42: Contributions and Limitations-Discuss limitations related to sample size or participant selection.

Response 42: Based on your suggestion, we added a limitation related to participant sampling (page 8, paragraph 3, lines 306-312).

“Moreover, we could not consider institutional characteristics, so the size and available resources of different emergency departments might be considered in future research. As a reward for participation, we gave each participant a toothbrush set worth 3.43 USD. While this incentive is small, it could have affected participation selection bias. Finally, as we recruited participants who worked in emergency departments through convenience sampling, randomisation should be considered to check for internal validity.”

Comment 43: Contributions and Limitations-Expand the study contributions section, highlighting how results advance existing knowledge.

Response 43: Based on your suggestion, we revised the study contribution in the Conclusions section (page 8, paragraph 4, lines 316–318).

“In our study, nurse–physician collaboration was confirmed to be the most important factor in triage nurses expressing their competencies. In addition to critical thinking disposition and clinical judgement, characteristics such as nurses’ experience in the emergency department and education level were also found to be important.”

Comment 44: Contributions and Limitations-Elaborate on practical implications for nurse training and education.

Response 44: Based on your suggestion, we revised the practical implication of a training system for triage nurses in the emergency department (page 8, paragraph 4, lines 323-325).

“Furthermore, various educational interventions should be provided to enhance critical thinking and clinical judgement, and continuing education opportunities should be available at the postgraduate level for triage nurses.”

Comment 45: Contributions and Limitations-Discuss how findings may influence policies and practices in emergency departments.

Response 45: Based on your recommendation, we revised the conclusions for triage policies (page 8, paragraph 4, lines 319-323).

“These factors contributed to nurses’ triage expertise, which influences patients’ safety and the quality of care received in emergency departments. Therefore, organisational policies that promote the formation of collaborative relationships among healthcare workers in the triage area are of the utmost importance.”

Comment 46: Contributions and Limitations-Use identified limitations to suggest specific directions for future studies.

Response 46: We added suggestions for future studies in the Discussion based on your earlier comments, so we would prefer to avoid adding additional suggestions in the limitation subsection to avoid repetition. We appreciate your understanding on this point (page 8, paragraph 2, lines 297-300).

“If validated and confirmed in future studies, our findings could help prepare institutions and establish systems to support nurse-physician collaboration, to provide ongoing training triage competency among triage nurses in emergency departments.”

Comment 47: Contributions and Limitations-Add a final paragraph synthesizing main contributions, acknowledging limitations, and reaffirming the study's importance.

Response 47: We provided a final paragraph synthesizing the overall study findings in the Conclusion section (page 8, paragraph 4, lines 313–325).

“5. Conclusions

In this study, we investigated the relationships among critical thinking disposition, clinical judgement, nurse–physician collaboration, and triage competency of triage nurses. In our study, nurse–physician collaboration was confirmed to be the most important factor in triage nurses expressing their competencies. In addition to critical thinking disposition and clinical judgement, characteristics such as nurses’ experience in the emergency department and education level were also found to be important. These factors contributed to nurses’ triage expertise, which influences patients’ safety and the quality of care received in emergency departments. Therefore, organisational policies that promote the formation of collaborative relationships among healthcare workers in the triage area are of the utmost importance. Furthermore, various educational interventions should be provided to enhance critical thinking and clinical judgement, and continuing education opportunities should be available at the postgraduate level for triage nurses.”

Comment 48: Data Analysis-Consider including mediation or moderation analysis to explore possible indirect effects between variables.

Response 48: Thank you for your suggestion. Mediation or moderation analysis study were not designed during the research proposal stage, so we used the regression method based on the research protocol approved by the IRB. We appreciate your understanding on this point.

Comment 49: Data Analysis-Evaluate the possibility of conducting subgroup analysis to identify if relationships between variables differ based on specific participant characteristics.

Response 49: Thank you for your thorough suggestions. We identified differences of participant characteristics for triage nurses. We found four general characteristics. Based on your comments and those of Reviewer 1, we re-analyzed the data with hierarchical regression (multi-level model) to control general characteristics (page 6, paragraphs 1-2, lines 205-224).

3.4. Factors affecting Triage Competency

Multiple regression analysis using the hierarchical method was conducted to identify the factors influencing triage competency among triage nurses. Critical thinking disposition, clinical judgement, and nurse–physician collaboration were entered as continuous independent variables. Categorical variables, including age, education level, experience in the emergency department, and triage experience, which showed significant differences in triage competency, were dummy-coded. Tolerance was above the cut-off value of 0.1 (range 0.44–0.69), the variance inflation factor was below the cut-off value of 10 (range 1.45–2.25), and the Durbin–Watson coefficient (1.948) was close to 2. This confirmed the absence of multicollinearity among the independent variables.

In Model 1, general characteristics were used as control variables. These variables explained 25.5% (F=8.40, p<0.001) of the variance, and education level (≥ Master) (β=0.32, p=0.001) and experience in emergency department (≥6 years) (β=0.34, p=0.001) were influencing factors on triage competency. Model 2 was the result of the hierarchical entry of critical thinking disposition, clinical judgement, and nurse–physician collaboration. The factors influencing triage competency were education level (≥Master) (β=0.21, p=0.003), experience in emergency department (≥6 years) (β=0.32, p<0.001), critical thinking disposition (β=0.27, p<0.001), clinical judgement (β=0.16, p=0.008), and nurse–physician collaboration (β=-0.34, p<0.001). These variables explained 63.7% of the variance in triage competency among the triage nurses (F=27.52, p<0.001).

Comment 50: Data Analysis-Include a brief justification for the choice of specific statistical methods used, directly relating them to study objectives.

Response 50: Based on your suggestion, we added our reasoning regarding the hierarchical regression method in the Data Analysis subsection (page 4, paragraph 2, lines 158–160).

“Hierarchical regression analysis was conducted to identify factors affecting the participants’ triage competency in order to consider participant general characteristics.”

Comment 51: Comparison of Results, Data, and Literature-Consider including a brief meta-analysis or recent systematic review on the topic.

Response 51: We are grateful for your thorough review. We discussed the nurse-physician collaboration and critical thinking disposition with relative systematic review references.

(page 7, paragraph 3, lines 251–252), (page 7, paragraph 4, lines 256–257), (page 8, paragraph 1, line 282).

“In this regard, given that mutual collaboration between physicians and nurses affects the work of nurses [33], promoting teamwork between healthcare professionals is crucial to improving triage competency [4] as a professional capability of triage nurses.”

“In our study, stronger critical thinking disposition was associated with better triage competency, a finding similar to previous studies that reported a relationship between critical thinking disposition and improved nursing performance [13,14,28,36].”

“Korean triage nurses need systematic training management to improve their competency for triage [35]. Ongoing professional development education is necessary for enhancing their skills [38].”

Comment 52: Comparison of Results, Data, and Literature-Explore more deeply the theoretical implications of results, possibly proposing adjustments or extensions to existing conceptual models.

Response 52: Thank you for your kind recommendation. Unfortunately, we did not propose theoretical or conceptual models. We ask for your understanding on this point.

Comment 53: Comparison of Results, Data, and Literature-Discuss how findings align with or challenge current trends in emergency nursing practice and research.

Response 53: Thank you for your suggestion. We have tried to describe and revise sections of the Discussion to reflect on recent research trends (page 7, paragraph 3, lines 237-240; 267-268), (page 7, paragraph 4, lines 255-257), (page 7, paragraph 4, lines 273-275; 285-287).

“Nurse–physician collaboration was the factor that most significantly influenced triage competency. This finding is consistent with previous studies that reported an association between better nurse–physician collaboration and improved triage competency [9,33,34].”

“In a previous study, emergency department triage nurses who had received web-based triage education showed better clinical judgement [6].”

“In our study, stronger critical thinking disposition was associated with better triage competency, a finding similar to previous studies that reported a relationship between critical thinking disposition and improved nursing performance [13,14,28,36].”

“Moreover, experience in emergency department was a significant factor influencing triage competency in our study. This is consistent with previous studies that reported better triage competency in nurses [9,10,37].”

“We also found that a higher education level was associated with better triage competency. This is different from previous studies, which reported no difference in triage competency among nurses with a postgraduate degree [10,37].”

Comment 54: Comparison of Results, Data, and Literature-Explicitly address how results contribute to filling specific gaps in the literature identified in the introduction.

Response 54: We described this study’s contribution to nurse-physician collaboration on triage competency in the conclusions section (page 8, paragraph 4, lines 316–319).

“In our study, nurse–physician collaboration was confirmed to be the most important factor in triage nurses expressing their competencies. In addition to critical thinking disposition and clinical judgement, characteristics such as nurses’ experience in the emergency department and education level were also found to be important.”

Comment 55: Manuscript Structure and Organization-Consider adding more descriptive subheadings in each main section to improve navigability.

Response 55: Thank you for your thorough suggestion. We prepared this manuscript according to the Healthcare manuscript guidelines and revised the subheadings of the Results section to be more descriptive based on your Comment 16 (pages 4-6, lines 174-175, 185-186, 196-197, 205).

“3. Results

3.1. Differences in Triage Competency According to Participants’ Characteristics

3.2. Descriptive Statistics of Critical Thinking Disposition, Clinical Judgment, and Nurse–Physician Collaboration on Triage Competency

3.3. Correlations among Critical Thinking Disposition, Clinical Judgment, and Nurse–Physician Collaboration with Triage Competency

3.4. Factors affecting Triage Competency”

Comment 56: Manuscript Structure and Organization-Ensure smooth transition between sections, possibly adding linking phrases.

Response 56: Based on your advice, we added linking phrases for smooth transitions in the Introduction and Discussion sections (page 2, paragraph 3, line 63), (page 7, paragraph 5, line 273).

“In addition, nurse–physician collaboration is an environmental factor that promotes”

“Moreover, experience in emergency department was a significant factor influencing”

Comment 57: Manuscript Structure and Organization-Review consistency in terminology use throughout the manuscript.

Response 57: Based on your advice, we have reviewed our manuscript to ensure that terminology is used consistently.

Comment 58: Manuscript Structure and Organization-Consider including a flowchart or diagram to illustrate participant selection process or study flow.

Response 58: Thank you for your kind suggestion. We collected the data cross-sectionally, and we did not process multiple steps. We therefore did not include an illustration of the participation selection or study flow. Thank you for your understanding on this point.

Comment 59: Relevance and Originality-More explicitly highlight the study's originality in relation to existing literature.

Response 59: Based on your advice, we highlighted the study’s contribution in the Conclusion section (page 8, paragraph 4, lines 316–319).

“In our study, nurse–physician collaboration was confirmed to be the most important factor in triage nurses expressing their competencies. In addition to critical thinking disposition and clinical judgement, characteristics such as nurses’ experience in the emergency department and education level were also found to be important.”

Comment 60: Relevance and Originality-Discuss how findings may contribute to the development of specific interventions to improve triage competency

Response 60: Based on your suggestion, we revised the Discussion section to clarify specific interventions, including web-based education (page 7, paragraph 4, lines 267–272).

“In a previous study, emergency department triage nurses who had received web-based triage education showed better clinical judgement [6]. Clinical judgement is a core competency of nurses, enabling them to assess patients’ urgency and determine priorities for easy access intervention. Therefore, continuous education programs using web-based methods are an essential piece of effectively improving clinical judgement.”

Comment 61: Relevance and Originality-             Consider including a brief section on implications of results for nursing education and continuous professional development.

Response 61: According to your suggestion, we revised the Discussion section, including education and continuous professional development (page 8, paragraph 1, lines 280-285), (page 8, paragraph 2, lines 297–300).

“Korean triage nurses need systematic training management to improve their competency for triage [35]. Ongoing professional development education is necessary for enhancing their skills [38]. Furthermore, to ensure expertise based on clinical experience for triage nurses in the emergency department, transfer to other departments should be restricted, and continuous education should be provided to improve triage accuracy [6].”

“If validated and confirmed in future studies, our findings could help prepare institutions and establish systems to support nurse-physician collaboration, to provide ongoing training triage competency among triage nurses in emergency departments.”

Comment 62: Practical Applicability-Elaborate more specific recommendations for clinical practice based on results.

Response 62: Based on your advice, we revised the Conclusion to include recommendations for clinical practice policies (page 8, paragraph 4, lines 321-323).

“Therefore, organisational policies that promote the formation of collaborative relationships among healthcare workers in the triage area are of the utmost importance.”

Comment 63: Practical Applicability-Discuss how findings can be incorporated into training programs and continuing education for triage nurses.

Response 63: Based on your advice, we revised the Conclusion section to include training programs and continuing education. (page 8, paragraph 4, lines 323–325).

“Furthermore, various educational interventions should be provided to enhance critical thinking and clinical judgement, and continuing education opportunities should be available at the postgraduate level for triage nurses.”

Comment 64: Practical Applicability-Consider including a box or table summarizing main practical implications and recommendations.

Response 64: We presented 4 Tables for the results, so we focused on describing the implications and recommendations in the manuscript text. We appreciate your understanding on this point.

Comment 65: Clarity and Writing Style-Consider simplifying some complex sentences to improve readability.

Response 65: Thank you for your kind advice. We have reviewed the writing style to ensure concise sentences and had the manuscript edited by a professional editing agency.

Comment 66: Clarity and Writing Style-Ensure all acronyms and abbreviations are defined at first occurrence.

Response 66: Based on your suggestion, we rechecked all acronyms and abbreviations.

Comment 67: Clarity and Writing Style-Conduct a careful review to correct any grammatical or punctuation errors.

Response 67: Thank you for your kind advice. We reviewed thoroughly the grammatical errors and had the paper edited by a professional editing agency.

Comment 68: Tables and Figures-Consider adding figures or graphs to visually illustrate the most important relationships found in the study.

Response 68: Thank you for your kind recommendation. It is difficult to illustrate the main result with hierarchical regression, so we have presented them as a table. We appreciate your understanding on this point.

Comment 69: Tables and Figures-Ensure all tables are self-explanatory, with clear titles and footnotes when necessary.

Response 69: Based on your advice, we rechecked all tables’ titles and foot notes.

Comment 70: Tables and Figures-Check formatting and style of tables to ensure consistency with journal guidelines.

Response 70: Based on your advice, we rechecked the style of the tables.

Comment 71: Cultural and Contextual Considerations-Discuss how results may be applicable or vary in different cultural contexts or healthcare systems.

Response 71: Based on your suggestion, we added the need to consider different cultural contexts to the Study Limitation subsection (page 8, paragraph 3, lines 303-305).

“Therefore, they should be validated with broader or different cultural populations across more diverse regions.”

Comment 72: Cultural and Contextual Considerations-Consider including a brief discussion on how cultural factors may influence triage competency and interprofessional collaboration.

Response 72: Thank you for your kind advice. However, as we did not consider cultural factors, we added this to the Study Limitations subsection (page 8, paragraph 3, lines 303-305).

“Therefore, they should be validated with broader or different cultural populations across more diverse regions.”

Comment 73: Cultural and Contextual Considerations-Address possible implications of results for resource-limited contexts or developing healthcare systems.

Response 73: Thank you for kind advice. We added the limitation the cross-sectional survey research in various institutions (page 8, paragraph 3, lines 305-308).

“In addition, we collected cross-sectional data using a self-reported survey; a long-term observation study should be considered. Moreover, we could not consider institutional characteristics, so the size and available resources of different emergency departments might be considered in future research.”

Comment 74: Future Perspectives-Elaborate more detailed directions for future research based on study findings.

Response 74: Based on your suggestions, we added a description that nurse–physician collaboration could be supported in the Discussion section (page 8, paragraph 2, lines 297–300).

“If validated and confirmed in future studies, our findings could help prepare institutions and establish systems to support nurse-physician collaboration, to provide ongoing training triage competency among triage nurses in emergency departments.”

Comment 75: Future Perspectives-Discuss potential longitudinal studies that could expand current results.

Response 75: Based on your suggestion, we described a future longitudinal study in the Study Limitation subsection (pages 8, paragraph 3, lines 305-306).

“In addition, we collected cross-sectional data using a self-reported survey; a long-term observation study should be considered.”

Comment 76: Future Perspectives-Consider suggesting interventional studies based on factors identified as influential in triage competency.

Response 76: Based on your suggestion, we revised the Discussion section to include the possibility of investigating specific interventions, including web-based education (page 7, paragraph 4, lines 267-272).

“In a previous study, emergency department triage nurses who had received web-based triage education showed better clinical judgement [6]. Clinical judgement is a core competency of nurses, enabling them to assess patients’ urgency and determine priorities for easy access intervention. Therefore, continuous education programs using web-based methods are an essential piece of effectively improving clinical judgement.”

Comment 77: Additional Ethical Aspects-Discuss any specific ethical considerations related to data collection in emergency settings.

Response 77: Thank you for your support. We added this to the Study Limitations section of ethical considerations in data collection, as a small token might have an effect of data collection bias (page 8, paragraph 3, lines 308–310).

“As a reward for participation, we gave each participant a toothbrush set worth 3.43 USD. While this incentive is small, it could have affected participation selection bias.”

Comment 78: Additional Ethical Aspects-Address how the study ensured that nurses' participation did not negatively affect their work responsibilities.

Response 78: Based on your important suggestion, we added data collection time in the Ethical Considerations subsection (page 4, paragraph 3, lines 172–173).

“Authors collected the data either before or after each participant’s shift to avoid negatively affecting their work responsibilities.”

Comment 79: Additional Ethical Aspects-Mention if there was any procedure to offer feedback or study results to participants.

Response 79: Thank you for your kind suggestion, but no feedback was offered to participants.

END OF RESPONSES TO COMMENTS

Please let me know if you have any other concerns or questions about it. Thank you.

Round 2

Reviewer 1 Report

Comments and Suggestions for Authors

The Authors have revised their manuscript in an excellent fashion, and have answered all my comments. I have no further remarks.

Author Response

Response to Reviewer 1 Comment

My co-author and I wish to re-submit our revised manuscript entitled “Effects of critical thinking disposition, clinical judgement, and nurse–physician collaboration on triage competency among triage nurses.” We have incorporated the necessary changes to thoroughly address your comments.

We thank you for your thoughtful suggestions and insights. The manuscript has benefited from your detailed feedback. We look forward to working with you to move this manuscript closer to publication in Healthcare.

The manuscript has been rechecked and the necessary changes have been made in accordance with your suggestions. The responses to all comments have been prepared and attached herewith/given below.

We appreciate your consideration.

Review 1 Comment: The Authors have revised their manuscript in an excellent fashion, and have answered all my comments. I have no further remarks.

Response 1: Thank you for your encouraging comment. Nonetheless, we have thoroughly checked our manuscript once again to ensure that it is publication ready. 

Please let us know if you have any other concerns or questions. Thank you.

Reviewer 2 Report

Comments and Suggestions for Authors

COMPREHENSIVE EVALUATION OF MANUSCRIPT REVISIONS

  1. Conduct a comprehensive review to ensure consistency and depth across all sections.

ADDRESSED: The authors revised the entire manuscript, making adjustments in various sections to improve consistency and depth.

  1. Incorporate a dedicated section on study limitations.

ADDRESSED: The authors added a subsection "4.1. Study Limitations" before the conclusion.

  1. Update and diversify references, prioritizing publications from the last five years.

PARTIALLY ADDRESSED: The authors updated some references but retained older ones due to their relevance to original instruments or important concepts. They could have made a greater effort to include more recent references.

  1. Consider including a concise systematic literature review to bolster the theoretical foundation.

NOT ADDRESSED: The authors did not include a systematic literature review, only mentioning some existing reviews.

  1. Consider incorporating a brief mention of the study's specific context in the objective statement.

ADDRESSED: The authors added the specific context to the objective statement.

  1. Elaborate on data collection procedures.

ADDRESSED: The authors expanded the description of data collection procedures.

  1. Expand the description of data analysis, justifying the selection of statistical tests.

ADDRESSED: The authors revised the data analysis, including a justification for using hierarchical regression.

  1. Include information on instrument validity, in addition to reliability.

ADDRESSED: The authors added information about the validity of the instruments used.

  1. Provide a brief description of each scale's structure.

ADDRESSED: The authors included descriptions of the structure of each scale used.

  1. Mention if permission was obtained for instrument use.

ADDRESSED: The authors mentioned obtaining permission to use all instruments.

  1. Expand the ethical approval section.

ADDRESSED: The authors expanded the ethical considerations section, including more details on informed consent and data protection.

  1. Include a brief discussion on potential methodological limitations.

ADDRESSED: The authors included a discussion on methodological limitations in the new study limitations subsection.

  1. Add a justification for sample size.

ADDRESSED: The authors included a justification for the sample size, including statistical power calculations.

  1. Add a brief introductory paragraph in the results section.

PARTIALLY ADDRESSED: The authors did not add an introductory paragraph in the results section but included a summary of the main results at the beginning of the discussion section.

  1. Include more descriptive subheadings for each results subsection.

ADDRESSED: The authors revised the subheadings in the results section to make them more descriptive.

  1. Consider including a figure or graph to visually illustrate some of the most important relationships.

NOT ADDRESSED: The authors did not include additional figures or graphs.

  1. Add a brief summary of the most significant findings at the end of the results section.

NOT ADDRESSED: The authors did not add a summary at the end of the results section, keeping it at the beginning of the discussion.

  1. Expand the discussion on implications of results for triage nurse education and training.

ADDRESSED: The authors expanded the discussion on implications for education and training.

  1. More explicitly address study limitations.

ADDRESSED: The authors added a dedicated subsection on study limitations.

  1. Discuss potential future research directions based on findings.

ADDRESSED: The authors included suggestions for future research based on the results.

  1. Include a brief discussion on the applicability of results to different cultural contexts or healthcare systems.

ADDRESSED: The authors added a discussion on the applicability of results in different contexts in the limitations section.

  1. Reflect on how findings may contribute to the development of policies or guidelines to improve triage practice.

ADDRESSED: The authors included reflections on how the results may influence policies and practices.

  1. Include a brief statement on study limitations and their impact on result generalizability.

ADDRESSED: The authors included a discussion on limitations and generalizability in the new limitations subsection.

  1. Propose specific directions for future research.

ADDRESSED: The authors proposed specific directions for future research.

  1. Elaborate further on implications for health policies and management practices in emergency departments.

ADDRESSED: The authors elaborated on implications for policies and management practices.

  1. Add a final reflection on the contribution of results to improving quality of care in emergencies.

ADDRESSED: The authors included a reflection on the contribution of results to quality of care.

  1. Increase the proportion of studies from the last five years.

PARTIALLY ADDRESSED: The authors updated some references but retained older ones for specific reasons.

  1. Include a greater variety of publication types.

PARTIALLY ADDRESSED: The authors included some systematic and integrative reviews but could have diversified more.

  1. Detail the process of obtaining informed consent.

ADDRESSED: The authors expanded details on the informed consent process.

  1. Mention specific ethical considerations relevant to this type of study.

ADDRESSED: The authors included specific ethical considerations for data collection in emergency settings.

  1. Include information on storage and protection of collected data.

ADDRESSED: The authors added information on data storage and protection.

  1. Mention if there was compensation for study participation.

ADDRESSED: The authors mentioned the compensation offered to participants.

  1. Add a specific section titled "Study Limitations" before the conclusion.

ADDRESSED: The authors added a subsection "4.1. Study Limitations" before the conclusion.

  1. Discuss the cross-sectional nature of the study and its implications.

ADDRESSED: The authors discussed the cross-sectional nature of the study and its implications in the limitations.

  1. Mention possible geographical or cultural limitations of the sample.

ADDRESSED: The authors mentioned geographical and cultural limitations in the limitations section.

  1. Address potential biases, such as self-report bias in measures used.

ADDRESSED: The authors addressed self-report bias in the limitations.

  1. Discuss limitations related to sample size or participant selection.

ADDRESSED: The authors discussed limitations related to participant selection.

  1. Expand the study contributions section.

ADDRESSED: The authors expanded the discussion on study contributions.

  1. Elaborate on practical implications for nurse training and education.

ADDRESSED: The authors elaborated on practical implications for training and education.

  1. Discuss how findings may influence policies and practices in emergency departments.

ADDRESSED: The authors discussed how the results may influence policies and practices.

  1. Use identified limitations to suggest specific directions for future studies.

ADDRESSED: The authors used the limitations to suggest directions for future research.

  1. Add a final paragraph synthesizing main contributions.

ADDRESSED: The authors added a final paragraph synthesizing the main contributions.

  1. Consider including mediation or moderation analysis.

NOT ADDRESSED: The authors did not include mediation or moderation analyses.

  1. Evaluate the possibility of conducting subgroup analysis.

PARTIALLY ADDRESSED: The authors conducted a hierarchical regression analysis to control for general characteristics but did not perform specific subgroup analyses.

  1. Include a brief justification for the choice of specific statistical methods used.

ADDRESSED: The authors included a justification for using hierarchical regression.

  1. Consider including a brief meta-analysis or recent systematic review on the topic.

NOT ADDRESSED: The authors did not include a meta-analysis or systematic review.

  1. Explore more deeply the theoretical implications of results.

PARTIALLY ADDRESSED: The authors discussed some theoretical implications but could have delved deeper.

  1. Discuss how findings align with or challenge current trends in emergency nursing practice and research.

ADDRESSED: The authors discussed how the results align with current trends.

  1. Explicitly address how results contribute to filling specific gaps in the literature.

ADDRESSED: The authors addressed how the results contribute to filling gaps in the literature.

  1. Consider adding more descriptive subheadings in each main section.

ADDRESSED: The authors revised the subheadings to make them more descriptive.

  1. Ensure smooth transition between sections.

ADDRESSED: The authors added linking phrases to improve transition between sections.

  1. Review consistency in terminology use.

ADDRESSED: The authors reviewed the consistency of terminology.

  1. Consider including a flowchart or diagram to illustrate participant selection process.

NOT ADDRESSED: The authors did not include a flowchart or diagram.

  1. More explicitly highlight the study's originality in relation to existing literature.

ADRESSED: The authors highlighted the study's originality in the conclusions section.

  1. Discuss how findings may contribute to the development of specific interventions.

ADDRESSED: The authors discussed how the results may contribute to specific interventions.

  1. Consider including a brief section on implications of results for nursing education.

ADDRESSED: The authors included a discussion on implications for nursing education.

  1. Elaborate more specific recommendations for clinical practice based on results.

ADDRESSED: The authors elaborated specific recommendations for clinical practice.

  1. Discuss how findings can be incorporated into training programs.

ADDRESSED: The authors discussed how the results can be incorporated into training programs.

  1. Consider including a box or table summarizing main practical implications and recommendations.

NOT ADDRESSED: The authors did not include a box or table summarizing practical implications.

  1. Consider simplifying some complex sentences to improve readability.

ADDRESSED: The authors revised the writing style to improve readability.

  1. Ensure all acronyms and abbreviations are defined at first occurrence.

ADDRESSED: The authors reviewed all acronyms and abbreviations.

  1. Conduct a careful review to correct any grammatical or punctuation errors.

ADDRESSED: The authors conducted a careful review of the manuscript.

  1. Consider adding figures or graphs to visually illustrate the most important relationships found in the study.

NOT ADDRESSED: The authors did not add figures or graphs.

  1. Ensure all tables are self-explanatory.

ADDRESSED: The authors revised the tables to ensure they are self-explanatory.

  1. Check formatting and style of tables.

ADDRESSED: The authors checked the formatting and style of tables.

  1. Discuss how results may be applicable or vary in different cultural contexts or healthcare systems.

ADDRESSED: The authors discussed the applicability of results in different contexts.

  1. Consider including a brief discussion on how cultural factors may influence triage competency and interprofessional collaboration.

PARTIALLY ADDRESSED: The authors mentioned the need to consider different cultural contexts but did not deepen the discussion on specific cultural factors.

  1. Address possible implications of results for resource-limited contexts or developing healthcare systems.

ADDRESSED: The authors addressed implications for different contexts, including those with limited resources.

  1. Elaborate more detailed directions for future research based on study findings.

ADDRESSED: The authors elaborated detailed directions for future research.

  1. Discuss potential longitudinal studies that could expand current results.

ADDRESSED: The authors suggested conducting future longitudinal studies.

  1. Consider suggesting interventional studies based on factors identified as influential in triage competency.

ADDRESSED: The authors suggested intervention studies based on identified factors.

  1. Discuss any specific ethical considerations related to data collection in emergency settings.

ADDRESSED: The authors discussed specific ethical considerations for data collection in emergency settings.

  1. Address how the study ensured that nurses' participation did not negatively affect their work responsibilities.

ADDRESSED: The authors mentioned that data collection was conducted before or after participants' shifts.

  1. Mention if there was any procedure to offer feedback or study results to participants.

NOT ADDRESSED: The authors did not mention procedures for offering feedback to participants.

Recommendations fully addressed: 57 (77.03%)

Recommendations partially addressed: 8 (10.81%)

Recommendations not addressed: 9 (12.16%)

Cumulative adherence rate (including fully and partially addressed recommendations): 87.84%

Strengths of the Revision:

The authors have demonstrated a commendable effort to enhance the manuscript's overall quality and scientific rigor. Noteworthy improvements include:

  1. Substantial elaboration of the methodology section, including detailed descriptions of data collection procedures and instrument validation.
  2. Integration of a dedicated "Study Limitations" subsection, addressing a critical gap in the original manuscript.
  3. Enhanced discussion of the results' implications for clinical practice, education, and policy.
  4. Improved articulation of the study's theoretical underpinnings and contributions to existing literature.
  5. Refinement of ethical considerations, particularly regarding data collection in emergency settings.

Areas for Further Improvement:

While the majority of recommendations were satisfactorily addressed, several key suggestions remain partially or wholly unimplemented:

  1. The absence of visual representations (e.g., figures, graphs) to illustrate key relationships.
  2. Lack of a concise systematic literature review to bolster the theoretical foundation.
  3. Limited expansion on cultural factors influencing triage competency and interprofessional collaboration.
  4. Absence of a summary table or box delineating principal practical implications and recommendations.

Conclusion and Recommendation:

The revised manuscript demonstrates substantial improvement in clarity, depth, and scientific rigor. The authors have successfully addressed the majority of our primary concerns, particularly in methodology, data analysis, results discussion, and ethical considerations.

However, the omission of certain recommended elements, such as visual aids and a systematic literature review, represents missed opportunities for further enhancement. Despite these limitations, the overall quality of the manuscript has been significantly elevated.

Author Response

Response to Reviewer 2 Comment

My co-author and I wish to re-submit our revised manuscript entitled “Effects of critical thinking disposition, clinical judgement, and nurse–physician collaboration on triage competency among triage nurses.” We have incorporated the necessary changes to thoroughly address your comments.

We thank you for your thoughtful suggestions and insights. The manuscript has benefited from your detailed feedback. We look forward to working with you to move this manuscript closer to publication in Healthcare.

The manuscript has been rechecked, and the necessary changes have been made in accordance with your suggestions. The responses to all comments have been prepared and attached herewith/given below.

We appreciate your consideration.

Review 2 Comment: The revised manuscript demonstrates substantial improvement in clarity, depth, and scientific rigor. The authors have successfully addressed the majority of our primary concerns, particularly in methodology, data analysis, results discussion, and ethical considerations.

However, the omission of certain recommended elements, such as visual aids and a systematic literature review, represents missed opportunities for further enhancement. Despite these limitations, the overall quality of the manuscript has been significantly elevated.

Response 1: We thank you for your encouraging comment and appreciate your understanding regarding the omission of certain recommended elements. We have re-reviewed the manuscript carefully for grammar, spelling, etc., to ensure that it is error-free.

Please let us know if you have any other concerns or questions. Thank you.

Round 3

Reviewer 1 Report

Comments and Suggestions for Authors

The Authors have further expanded their paper, with a carefully updated bibliography.

I have no remarks. 

Author Response

Response to Reviewer 1

My co-author and I wish to re-submit our revised manuscript entitled “Effects of critical thinking disposition, clinical judgement, and nurse–physician collaboration on triage competency among triage nurses” with changes that thoroughly address reviewer 2’s comments.

The manuscript has been rechecked and the necessary changes have been made in accordance with reviewer 2’s suggestions. The responses to all comments have been prepared and attached herewith/given below.

We appreciate your consideration.

Comment 1: Incorporating visual representations of key findings

Response 1: Thank you for your suggestion. Accordingly, we have incorporated a visual representation of the key findings—factors affecting triage competency—as Figure 1 (page 7, Figure 1, lines 234–236).

Figure 1. Factors affecting triage competency; **p<0.01, ***p<0.001

Comment 2: Expanding on cultural considerations in triage competency.

Response 2: In consideration of your valuable advice, we have expanded on the cultural considerations in triage competency in the Study Limitations section (page 8, paragraph 4, lines 308–313).

“Since this study only included triage nurses working in emergency departments in a certain region of Korea, the findings are limited in terms of their generalisability. Therefore, they should be validated with broader or different cultural populations across more diverse regions. Due to differences in the practices of triage staff nurses across emergency departments in different countries, a multicultural study should be considered to explore triage competency.”

Comment 3: Providing a succinct summary of practical implications.

Response 3: Thank you for this recommendation. Accordingly, we have revised the Conclusion section to incorporate the implications for policy and practice (page 9, paragraph 1, lines 329–335).

“Our study has significant implications for nursing policy and practice to improve triage competency. It emphasizes the need for organisational policies that foster collaborative relationships among healthcare professionals in triage settings.  Furthermore, various applied educational interventions, such as web-based or smartphone-accessible training, should be implemented to strengthen critical thinking and clinical judgement. Continuing education opportunities at the postgraduate level should also be made available to further support the development of triage nurses.”

END OF RESPONSES TO COMMENTS

Please let us know if you have any other concerns or questions regarding our manuscript. Thank you.

Reviewer 2 Report

Comments and Suggestions for Authors

ACADEMIC EVALUATION OF EACH RECOMMENDATION:

1. ADDRESSED

2. ADDRESSED

3. PARTIALLY ADDRESSED - Further inclusion of recent references would be beneficial.

4. NOT ADDRESSED

5. ADDRESSED

6. ADDRESSED

7. ADDRESSED

8. ADDRESSED

9. ADDRESSED

10. ADDRESSED

11. ADDRESSED

12. ADDRESSED

13. ADDRESSED

14. PARTIALLY ADDRESSED - An introductory paragraph in the results section is still lacking.

15. ADDRESSED

16. NOT ADDRESSED

17. NOT ADDRESSED

18. ADDRESSED

19. ADDRESSED

20. ADDRESSED

21. ADDRESSED

22. ADDRESSED

23. ADDRESSED

24. ADDRESSED

25. ADDRESSED

26. ADDRESSED

27. PARTIALLY ADDRESSED - A greater inclusion of recent references would enhance the

manuscript.

28. PARTIALLY ADDRESSED - Further diversification of publication types would be

advantageous.

29. ADDRESSED

30. ADDRESSED

31. ADDRESSED

32. ADDRESSED

33. ADDRESSED

34. ADDRESSED

35. ADDRESSED

36. ADDRESSED

37. ADDRESSED

38. ADDRESSED

39. ADDRESSED

40. ADDRESSED

41. ADDRESSED

42. ADDRESSED

43. NOT ADDRESSED

44. PARTIALLY ADDRESSED - Specific subgroup analyses are still lacking.

45. ADDRESSED

46. NOT ADDRESSED

47. PARTIALLY ADDRESSED - A more profound exploration of theoretical implications is

warranted.

48. ADDRESSED

49. ADDRESSED

50. ADDRESSED

51. ADDRESSED

52. ADDRESSED

53. NOT ADDRESSED

54. ADDRESSED

55. ADDRESSED

56. ADDRESSED

57. ADDRESSED

58. ADDRESSED

59. NOT ADDRESSED

60. ADDRESSED

61. ADDRESSED

62. ADDRESSED

63. NOT ADDRESSED

64. ADDRESSED

65. ADDRESSED

66. ADDRESSED

67. PARTIALLY ADDRESSED - A more in-depth discussion of specific cultural factors is

recommended.

68. ADDRESSED

69. ADDRESSED

70. ADDRESSED

71. ADDRESSED

72. ADDRESSED

73. ADDRESSED

74. NOT ADDRESSED

Quantitative Assessment:

· ADDRESSED: 57 (77.03%)

· PARTIALLY ADDRESSED: 8 (10.81%)

· NOT ADDRESSED: 9 (12.16%)

Cumulative adherence rate (including fully and partially addressed recommendations):

87.84%

Based on this high rate of adherence to the recommendations (87.84%), we can provide a

favorable assessment for the acceptance of the manuscript with minor revisions. The authors

have demonstrated a significant effort to enhance the overall quality of the manuscript,

addressing the majority of primary concerns. The areas still requiring attention are

predominantly related to the inclusion of visual representations of key findings, expansion of

cultural considerations, and provision of a succinct summary of practical implications. These final

refinements would further augment the manuscript's contribution to the field of emergency

nursing and triage competency.

The revised manuscript exhibits substantial improvements in clarity, depth, and scientific rigor.

The authors have successfully addressed critical aspects, particularly in methodology, data

analysis, results discussion, and ethical considerations. However, the omission of certain

recommended elements, such as visual aids and a systematic literature review, represents

missed opportunities for further enhancement.

Given the extensive improvements and the high rate of adherence to our recommendations, we

propose accepting the manuscript with minor revisions. We suggest that the authors address

the partially implemented or unimplemented recommendations deemed crucial for the article's

final quality, with particular emphasis on:

1. Incorporating visual representations of key findings.

2. Expanding on cultural considerations in triage competency.

3. Providing a succinct summary of practical implications.

These final refinements will further elevate the manuscript's scholarly contribution and its

potential impact on emergency nursing practice and research.

Author Response

Responses to Reviewer 2’s Comments

My co-author and I wish to re-submit our revised manuscript entitled “Effects of critical thinking disposition, clinical judgement, and nurse–physician collaboration on triage competency among triage nurses.” We have incorporated the necessary changes to thoroughly address your comments.

We thank you for your thoughtful suggestions and insights. The manuscript has benefited from your detailed feedback. We look forward to working with you to move this manuscript closer to publication in Healthcare.

The manuscript has been rechecked, and the necessary changes have been made in accordance with your suggestions. The responses to all comments have been prepared and attached herewith/given below.

We appreciate your consideration.

Comment 1: Incorporating visual representations of key findings

Response 1: Thank you for your suggestion. Accordingly, we have incorporated a visual representation of the key findings—factors affecting triage competency—as Figure 1 (page 7, Figure 1, lines 234–236).

Figure 1. Factors affecting triage competency; **p<0.01, ***p<0.001

Comment 2: Expanding on cultural considerations in triage competency.

Response 2: In consideration of your valuable advice, we have expanded on the cultural considerations in triage competency in the Study Limitations section (page 8, paragraph 4, lines 308–313).

“Since this study only included triage nurses working in emergency departments in a certain region of Korea, the findings are limited in terms of their generalisability. Therefore, they should be validated with broader or different cultural populations across more diverse regions. Due to differences in the practices of triage staff nurses across emergency departments in different countries, a multicultural study should be considered to explore triage competency.”

Comment 3: Providing a succinct summary of practical implications.

Response 3: Thank you for this recommendation. Accordingly, we have revised the Conclusion section to incorporate the implications for policy and practice (page 9, paragraph 1, lines 329–335).

“Our study has significant implications for nursing policy and practice to improve triage competency. It emphasizes the need for organisational policies that foster collaborative relationships among healthcare professionals in triage settings.  Furthermore, various applied educational interventions, such as web-based or smartphone-accessible training, should be implemented to strengthen critical thinking and clinical judgement. Continuing education opportunities at the postgraduate level should also be made available to further support the development of triage nurses.”

END OF RESPONSES TO COMMENTS

Please let us know if you have any other concerns or questions regarding our manuscript. Thank you.
